# The role of gene duplication and paralog specialisation in the evolution of the mammalian PRPS complex

Bibek R. Karki [1], Austin C. MacMillan [1], Sara Vicente-Muñoz[2], Kenneth D. Greis [1], Lindsey E. Romick [2] & John T. Cunningham [1] ✉

The phosphoribosyl pyrophosphate synthetase (PRPS) enzyme catalyzes a chokepoint reaction in nucleotide production, making it essential for life. Here, we show that the presence of multiple PRPS-encoding genes is a hallmark trait of eukaryotes, and we find that gains or losses of paralogs are associated with major branching events in the eukaryotic tree. We pinpoint the evolutionary origins and define the individual roles for each of the mammalian PRPS paralogs, which we demonstrate work together as a heterogeneous multicomponent complex. Employing isogenic cells representing all viable individual or combinatorial assembly states, we dissect the basic organizational principles of the enzyme complex and characterize the emergent properties responsible for paralog specialization, including new modes of regulation that govern complex assembly and activity in vivo. Collectively, our study demonstrates how evolution has transformed a single PRPS enzyme into a biochemical complex endowed with novel functional and regulatory features that fine-tune mammalian metabolism.

Phosphoribosyl pyrophosphate synthetase (PRPS) is an enzyme conserved across all forms of life, tracing back to the last universal common ancestor[1–3]. PRPS catalyzes the rate-limiting step in converting ribose-5-phosphate (R5P) to phosphoribosyl pyrophosphate (PRPP), a crucial precursor in the biosynthesis of nucleotides, amino acids, and lipids[4,5]. PRPS enzymes are broadly categorized into three classes: Class I[5–8] (the classical PRPS), found in nearly all eukaryotes, is allosterically regulated by downstream nucleotides−ADP/GDP (inhibitors) and inorganic phosphate (activator); Class II[9–11], present across eukaryotes, is devoid of allosteric modulation; and Class III[12–15], specific to Archaea, is activated by inorganic phosphate but insensitive to downstream nucleotides. Mammals exclusively possess three Class I PRPS isozymes−PRPS1, PRPS2, and PRPS3, which form a heteromeric complex with two additional homologs−PRPS-associated protein 1 (PRPSAP1) and PRPS-associated protein 2 (PRPSAP2)[16–22]. However, their evolutionary origins, the functional significance of these heteromeric assemblies and the specific roles of each homolog are still unclear.

Here, we perform evolutionary analyses to investigate the origins of mammalian PRPS homologs and employ biochemical and genetic approaches to establish the structure-function relationships that influence assembly and activity of the mammalian PRPS complex. While we find no evidence for archaeal Class III PRPS enzymes in eukaryotes, the presence of homologous Class II PRPS enzymes in Excavata, Diaphoretickes, and Amorphea lineages indicates orthology dating back to the last eukaryotic common ancestor (LECA). Coupling this evidence with the even greater prevalence of Class I PRPS enzymes in these three major eukaryotic branches, we conclude that multiple bacterial species likely contributed these key paralogous metabolic enzymes to LECA's biochemical framework (Supplementary Data 1 and 2). We sought to understand how enzymes of different classes cooperate and how respective evolutionary trajectories shaped PRPS enzyme function in modern day eukaryotes. Focusing on the mammalian Class I PRPS enzymes, we trace the origin of PRPSAP2 to a gene duplication of PRPS1 in the ancestor of all animals and fungi.

[1]Department of Cancer Biology, University of Cincinnati College of Medicine, Cincinnati, OH, USA. [2]Division of Pathology and Laboratory Medicine, Cincinnati Children's Hospital Medical Center, Cincinnati, OH, USA. ✉e-mail: cunnijn@ucmail.uc.edu

Additionally, we identify a second gene duplication event in the ancestor of jawed vertebrates that produced PRPS2 and PRPSAP1, from PRPS1 and PRPSAP2, respectively. We demonstrate that the hetero-meric PRPS enzyme complex is among the largest assemblies in mammalian cells, with different tissues achieving unique architectures by potentially varying the stoichiometric expression of individual components. We show the critical importance of proper PRPS complex assembly for enzyme functionality within cells as aberrant assembly perturbs global metabolic flux and decreases cellular fitness. Addi-tionally, we utilize genetic engineering to define the organization and assembly of the complex, establish preferential interactions among its members, and uncover regulatory mechanisms linking translational control to complex assembly. Our thorough phylogenomic analyses reveal multiple convergent evolutionary patterns giving rise to PRPSAP-like homologs throughout Amorphea, reinforcing the impor-tance of PRPS enzymes operating as multimeric complexes in diverse eukaryotes.

## Results

### Loss of Class II PRPS correlates with expansion of Class I PRPS encoding genes in eukaryotes

Bacteria and Archaea species typically express a single PRPS enzyme[6,7,12–15,23,24], whereas the few eukaryotic organisms studied to date possess multiple PRPS homologs[10,16,21,25,26]. We wondered whether this observation was extensible over the entire domain of eukaryotes. To address this, we curated an extensive catalog of PRPS enzyme sequences that served as a framework for interrogating how selective pressures contribute to the emergence of new properties over evolu-tionary timescales. Sequence-based homology searches and genomic analyses, including more than 100 newly annotated sequences con-firmed that Class I PRPS are the most prevalent class in eukaryotes, with many species' genomes encoding multiple Class I PRPS genes[27–29] (Fig. 1A and Supplementary Data 1). We also show the distribution of PRPS homologs in a recent alternative phylogenetic tree based on mitochondrial proteins of alphaproteobacterial origin, which projects two eukaryotic supergroups[30] (Supplementary Fig. 1A). We hypothe-sized that this increased repertoire of PRPS homologs imbues eukar-yotes with enhanced metabolic adaptability by virtue of additional regulatability and increased biosynthetic capacity.

Given the well-established evolutionary trajectory from opistho-konts to mammals[31], supported by relatively complete molecular phylogenetic data, we used mammals, that harbor five PRPS homologs, as a model system to investigate the evolutionary origins and func-tional significance of these multiple homologs. In addition to the three isozymes–PRPS1, PRPS2, and PRPS1L1[16,17]–mammals also possess PRPSAP1 and PRPSAP2[18,19], which feature insertions in the catalytic flexible (CF) loop (referred to as non-homologous regions (NHRs)) compared to the sequences found in PRPS isozymes. Interestingly, this pattern of CF loop insertions is not exclusive to Holozoa but is also found in Holomycota (for example, *Saccharomyces cerevisiae* Prs1 and Prs5 homologs[26]). PRPSAP2 orthologs (termed Prs1 in Holomycota) emerged in basal opisthokonts and share greater amino acid sequence identity with opisthokont PRPS1 than with PRPS proteins from non-opisthokont lineages (Supplementary Fig. 1B). Notably, Prs5 orthologs, which contain insertions in the regulatory flexible (RF) loop, arose in the ancestor of Holomycota and are exclusive to this lineage (Fig. 1A). Interestingly, some Prs5 orthologs also possess expanded CF loops in addition to RF loop. Comparative gene structure analysis revealed that orthologs of opisthokont PRPS1, PRPSAP2, and Prs5 share a conserved splice site junction, despite over a billion years of divergent evolution, suggesting that a gene duplication event in the ancestral PRPS1 encoding gene led to the emergence of PRPSAP2 and Prs5 (Fig. 1B and Supplementary Data 3). This data identifies two of the earliest known genetic events in Opisthokonta—one that occurred in the ancestor of all opisthokonts and another that distinguishes Holomycota from

Holozoa, thus showcasing the power of studying paralogous gene evolution.

Later in Holozoa evolution, another gene duplication occurred in the ancestor of jawed vertebrates, giving rise to PRPS2 and PRPSAP1 from PRPS1 and PRPSAP2, respectively (based on amino acid sequence homology summarized in Fig. 1C). Comparative analysis of exon-intron gene structures of PRPS1 with PRPS2 and PRPSAP2 with PRPSAP1 revealed conserved splice site junctions, thus confirming their origins (Fig. 1D, E). Synteny analyses reveal that post-duplication genes near *prps1* and *prpsap2* have corresponding paralogs near *prps2* and *prpsap1*, respectively (Supplementary Figs. 2A, B and 3A, B). This pat-tern rules out a single gene duplication event and instead suggests that a chromosomal segment was duplicated likely during the second whole genome duplication event (referred to as the 2R event), which took place after the emergence of urochordates but before the radiation of jawed vertebrates[32,33]. This co-evolution pattern between PRPS and PRPSAP, indicates a potential interdependence. Additionally, PRPS1L1, a testis-restricted intronless isoform[17], arose in the common ancestor of Eutherians presumably through a retrotransposition event involving PRPS1-encoding transcript (Fig. 1C). Although the high sequence identity of PRPS1L1 among Eutherians suggests selective pressure, the physiological role of testes-restricted PRPS1L1 remains unknown.

We next sought to explore the evolutionary patterns that fostered the expansion of Class I PRPS. In our initial survey of PRPS homologs across eukaryotes, we observed that Class II PRPS exhibits a sporadic distribution; however, its presence in Excavata, Diaphoretickes, and Amorphea along with the homology between these groups, suggests an origin dating back to the LECA. After our thorough characterization of Class I PRPS homologs in opisthokonts, we noticed a coevolutionary pattern between Class II and Class I PRPS homologs. Ancestral opis-thokonts and amoebozoans possessed Class II enzymes, but holomy-cotans lost them, concomitantly expanding their class I PRPS repertoire (Fig. 1A and Supplementary Data 2). Conversely, most holozoans retained Class II PRPS but possess only two Class I PRPS homologs. Of note, among the only three identified Holomycota spe-cies harboring Class II PRPS, genomic data from two indicates they are intronless, suggesting acquisition through horizontal transfer (Sup-plementary Fig. 4A and Supplementary Data 2). However, conserved splice site junctions in Class II PRPS from holozoans and amoebozoans suggest that each originated from a common ancestral source within their respective lineages (Supplementary Fig. 4B, C).

As in the case of Holomycota, loss of Class II PRPS in vertebrates coincided with the emergence of PRPS2 and AP1 isoforms in jawed vertebrates (Fig. 1C). These multiple examples of PRPS paralogs' loss and gain likely reflect compensatory selection, whereby the duplica-tion of Class I PRPS homologs offsets loss of the Class II PRPS, while simultaneously providing fertile templates for evolutionary innovations.

### Convergent evolution of PRPSAP-like homologs throughout Amorphea

Next, we investigated the functional innovations that accompanied the expansion of the Class I PRPS repertoire. In mammals, while PRPS1 and PRPS2 have been extensively studied as standalone enzymes in vitro, PRPSAPs (PRPSAP1 and PRPSAP2) have not been thoroughly characterized[6,8,23,34–38]. Motivated by the broad conservation of PRPSAP2 across opisthokonts, we investigated whether PRPSAPs could independently catalyze reactions similar to PRPS isozymes. In PRPS isozymes, two subunits of a dimer form a minimal functional unit to generate the active site, where the catalysis is facilitated by sur-rounding flexible loop regions[6,8]. The FLAG region of one PRPS monomer (Subunit A) coordinates ATP binding while the RF loop, pyrophosphate (PP) loop, CF loop, and R5P loop from another monomer (Subunit B) participate in catalysis (Fig. 2A). A comparison of

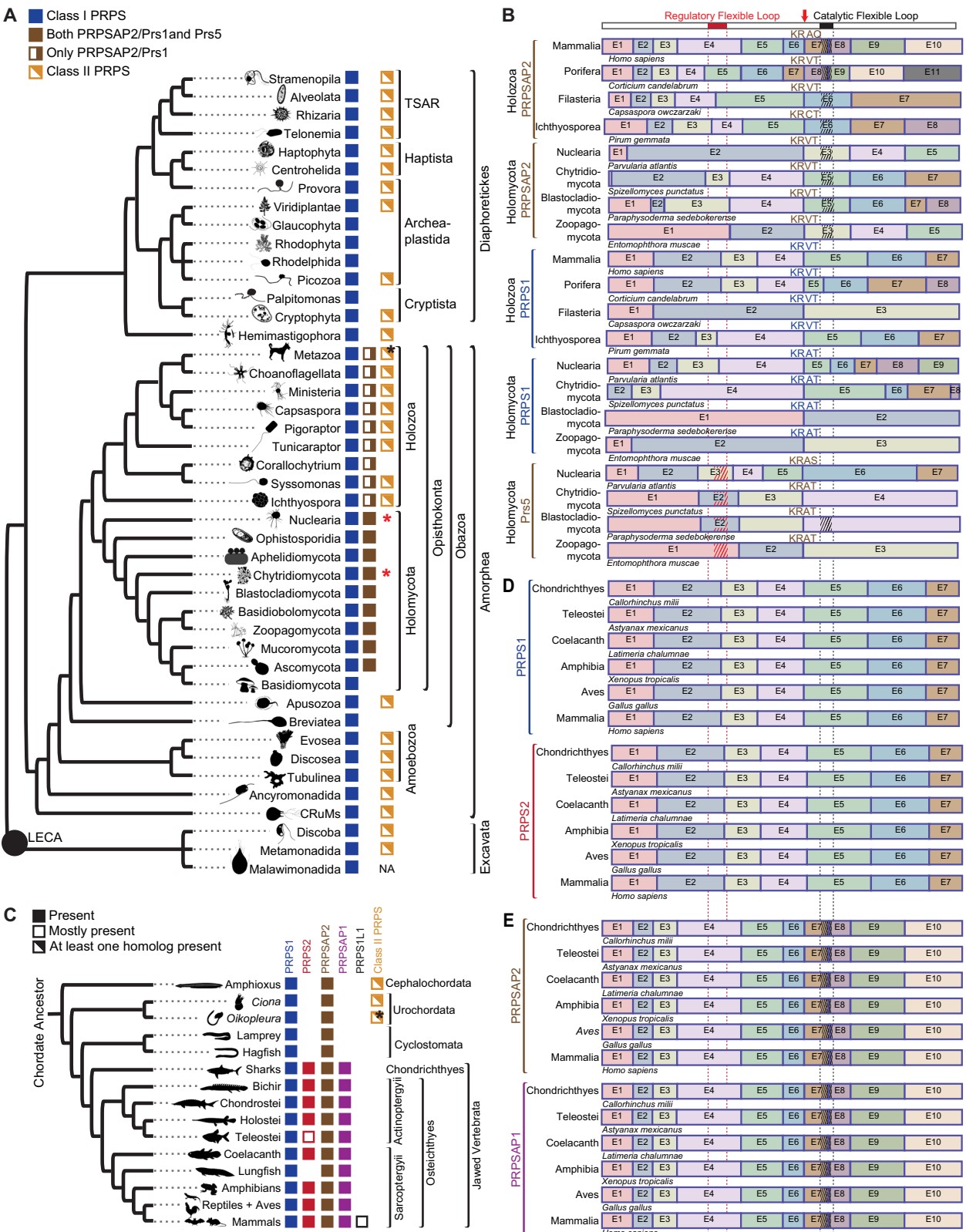

active site residues between PRPS1 and PRPSAP2 revealed a high degree of conservation in PRPS1 across Opisthokonta (Fig. 2B). In contrast, adaptive changes occurred rapidly in the loop regions of PRPSAP2 post-duplication suggesting strong selective pressure against the catalytic function. For instance, in human PRPS1, critical catalytic residues such as D171, K194, R196, N200, T225 have S177, G200, A202, E206, and D262 at corresponding positions in human PRPSAP2 (Fig.

2A and C), preventing PRPSAP2 from coordinating interactions between ATP and R5P in the closed conformation of catalytic loop, and from stabilizing the transition state[8,35,39,40]. Notably, the two interfaces that facilitate intramolecular PRPS subunit interactions[6,8,35,36,41]—a bent dimer essential for catalysis, and a parallel dimer required for allostery—are highly conserved in PRPSAPs suggesting a potential regulatory role through intermolecular binding with isozymes (Fig. 2D).

**Fig. 1 | Loss of Class II PRPS associated with expanded Class I PRPS homolog repertoire. A** Phylogenetic distribution of PRPS homologs in eukaryotes, with presence/absence of NHR-containing PRPS homologs in opisthokonts. PRPSAP2 denotes orthologs of mammalian PRPSAP2; Prs1 and Prs5 represent *S. cerevisiae* Prs1 and Prs5, respectively. Black asterisk indicates Class II presence across most metazoans, excluding Craniata. Red asterisks indicate Class II PRPS identified in only two Nuclearia species and one Chytridomycota species. **B** Conserved splice site junctions among PRPS homologs across different representative organisms in Opisthokonta. Gene structures for PRPS, PRPSAP2, and Prs5-encoding genes shown with exons as colored boxes; introns not displayed. Multiple sequence alignment (MSA) of translated sequences via Clustal Omega highlights a conserved splice site junction (red arrow) with adjacent amino acids shown. Top bar shows full *H. sapiens* PRPS1 with RF loop (red box) and CF loop (black box). Dotted red and black lines

project corresponding RF and CF loops positions, respectively onto other homologs. Insertions in RF and CF loops of PRPSAP2 and Prs5 shown with red and black hatch marks, respectively (hatch marks not to scale; NHRs vary in length). **C** Phylogenetic distribution of PRPS homologs in chordates. Mostly present – indicates present in most taxa. Black asterisk highlights that Class II PRPS is found in most urochordates, except *Oikopleura*, which has additional Class I PRPS homologs not observed in other organisms from this clade. **D, E** Conserved splice site junctions between PRPS1 and PRPS2 (**D**), and PRPSAP2 and PRPSAP1 (**E**) across different representative organisms in jawed Vertebrata. Gene structures for PRPS, PRPSAP2, and Prs5-encoding genes shown with exons as colored boxes; introns not displayed. Dotted red and black lines denote RF and CF loop regions, respectively; insertions in PRPSAP2 and PRPSAP1 shown as red and black hatch marks, respectively (hatch marks not to scale; NHRs vary in length).

We also identified duplicated Class I PRPS paralogs in certain species from Amoebozoa, Apusozoa, and CRuMs, characterized by insertions in their CF loop that distinguish them from their PRPS enzyme counterpart (Fig. 2E and Supplementary Fig. 4D). Curious whether the NHR-containing homologs in these Amorphean lineages originated from a common ancestor, we traced their evolutionary origins. Our analysis revealed evidence for independent origins through gene duplication events in Opisthokonta, Amoebozoa, Apusozoa, and CRuMs. For example, PrsB orthologs from Evosea exhibit greater amino acid sequence identity and share a conserved splice site junction with Evosea PrsA (PRPS), supporting their independent origins and providing strong evidence of convergent evolution (Supplementary Fig. 4E, F). Interestingly, these paralogs with CF loop expansion also exhibit poor conservation of active site residues, and conserved dimer interfaces−features bearing striking similarity to Opisthokonta PRPSAPs (Supplementary Fig. 5A–D and Supplementary Table 1). Collectively, the independent emergence of these homologs within Amorphea, with shared conservation and divergence patterns, strongly supports the case for convergent evolution with possible regulatory roles via interactions within a heteromeric PRPS complex.

## Mammalian PRPS enzyme operates as a multimeric complex that can be arranged in heterogeneous configurations

Given the conserved interaction interfaces between PRPS paralogs in mammalian cells, we experimentally tested whether they form a complex using mouse embryonic fibroblasts (NIH3T3) and human embryonic kidney cells (HEK293T). GFP-tagging all individual paralogs followed by immunoprecipitation (IP) assays revealed interactions among PRPS1, PRPS2, PRPSAP1 (AP1), and PRPSAP2 (AP2), confirming previous studies[20–22], and establishing the stable nature of the PRPS enzyme complex (Fig. 3A, B, Supplementary Fig. 6A–C, and Supplementary Data 4 and 5). A knock-in NIH3T3 cell line with ALFA-tagged PRPS1 confirmed these findings at endogenous expression levels as well (Supplementary Fig. 6D, E).

To further characterize mammalian (Class I) PRPS complex, we employed analytical size-exclusion chromatography (SEC) to assess its molecular weight. To establish intra-run controls suitable for cross sample comparison, we developed a panel of internal standards comprised of ubiquitously expressed proteins of known native molecular weights (MWs) and validated antibodies to serve as molecular weight markers for the fractions collected (see Methods). These standards cover a wide range of MWs from 1.5 MDa to 27 kDa (smaller than monomeric PRPS1). Nearly all PRPS paralogs are involved in heteromeric associations as evident from the overlapping retention times with an estimated average complex size of ~1.5 MDa for both NIH3T3 and HEK293T cells (Fig. 3C and Supplementary Fig. 6F). In contrast, exogenously expressed Class II PRPS in HEK293T cells revealed that Class II homologs, which exhibit poorly conserved dimer interfaces[5]− both bent and parallel−cannot interact with Class I homologs to assemble into a HMW complex and instead form smaller assemblies,

ranging from dimers to hexamers (Supplementary Fig. 6G). Based on this, we speculate that the capacity to form a heterogenous complex is unique to Class I. This is plausible as our evolutionary analysis is rife with examples of Class I duplication, while Class II duplications remain relatively rare. Moreover, the distinct features of Class I and Class II homologs suggest separate bacterial origins and argue against direct cooperativity between the two classes.

Given the substantial size of mammalian Class I PRPS complex, we wondered how the PRPS complex compared with other high molecular weight (HMW) assemblies of similar range. A proteomic analysis of HMW protein fractions from SEC (Supplementary Fig. 6H) identified a total of 262 unique proteins, which included ribosomal proteins and CAD (one of our standards), confirming enrichment for HMW proteins. This unbiased proteomic strategy revealed that among the eight cytosolic enzymes residing in HMW range, two were PRPS isozymes, indicating that the PRPS enzyme complex is one of the largest cytosolic metabolic assemblies in mammalian cells (Supplementary Fig. 6I, J and Supplementary Data 6).

We next explored whether the mammalian PRPS complex exists in a similar configuration in tissues compared to our proliferating cells in culture. PRPS complex components are ubiquitously expressed across rat tissues (PRPS1, PRPS2, and AP1)[22,42] and all complex members are present in human tissues as well (Supplementary Fig. 6K). However, the expression level of each component varies among tissues suggesting possible tissue-specific catalytic or regulatory roles. In the liver, the PRPS complex size was ~1.5 MDa, with all members interacting together (Fig. 3D). In the kidney, multiple configurations were observed: one complex at 1 MDa with PRPS1, AP1, and AP2, and another smaller complex consisting of PRPS1, PRPS2, and AP2, which may represent cell type differences in composition (Fig. 3E). In the lung, which exhibited low AP1 and AP2 expression, the complex size was the smallest among the tested tissues (Fig. 3F). These results demonstrate a heterogeneous array of PRPS complex configurations, capable of forming assemblies smaller than 100 kDa and greater than a megadalton. Importantly, variations in the architecture of the PRPS complex may be linked to the stoichiometric expression of PRPSAPs, and we speculate that PRPSAPs play a critical role in coordinating the assembly of the complex. Yeast genetics have demonstrated that interactions between PRPSAP-like orthologs and PRPS isozymes are essential for viability, highlighting the importance of maintaining proper PRPS complex architecture in other organisms[43,44].

## Genetic knockout studies reveal severe impact on metabolism and proliferation in cells exclusively expressing PRPS1

Previous attempts to characterize the components of the PRPS complex relied on crude protein purifications and in vitro functional assays. To address their role in cells, we sought to ascertain whether interactions between mammalian PRPS paralogs are functionally relevant in cells. To do so, we employed a CRISPR-Cas9 knockout (KO) strategy in NIH3T3 cells to generate all viable genetic knockout

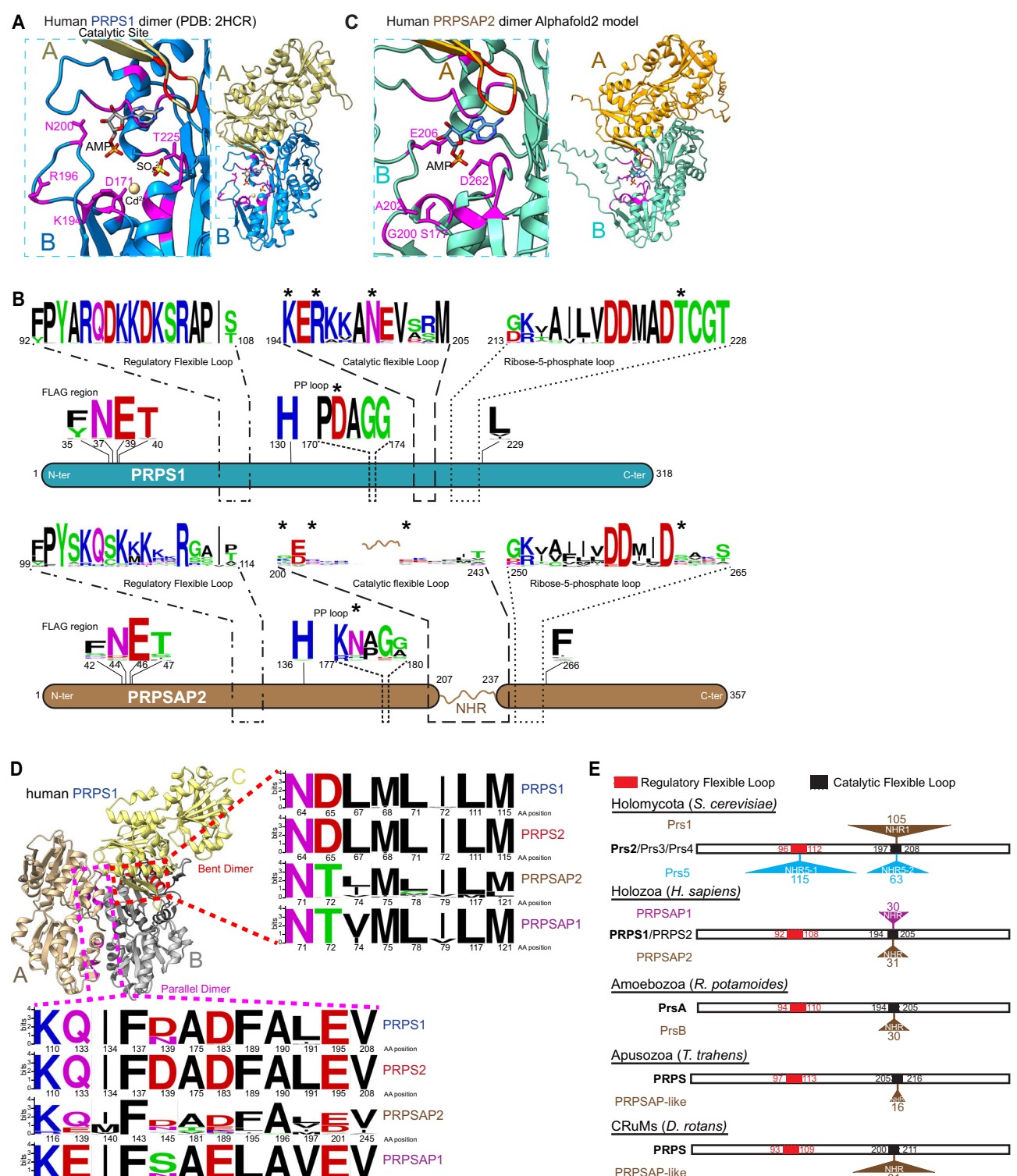

combinations (Fig. 4A and Supplementary Fig. 7A). Notably, we were unable to generate P1/AP1/AP2 knockouts, indicating that PRPS2 may not be sufficiently stable or active as a standalone enzyme in cells. Interestingly, the loss of AP1, AP2, or both associated proteins resulted in more severe cellular proliferation defects compared to the knockout of either PRPS1 or PRPS2 (ANOVA: $F$ (10, 22) = 104.3, $p < 0.0001$, $\eta^2 = 0.98$, 95% CI [0.97, 0.99]; Tukey's HSD comparing 60 h cell counts: P1 KO 1 vs AP1 KO 1, $p < 0.0001$; P1 KO 1 vs AP2 KO 1, $p < 0.0001$; P1 KO 1 vs AP1/AP2 KO 1, $p < 0.0001$; P2 KO 1 vs AP1 KO 1, $p = 0.0004$; P2 KO 1 vs AP2 KO 1, $p < 0.0001$; P2 KO 1 vs AP1/AP2 KO 1, $p < 0.0001$−Fig. 4B and

Supplementary Fig. 7B and ANOVA: $F$ (10, 32) = 136.0, $p < 0.0001$, $\eta^2 = 0.98$, 95% CI [0.98, 1.00]; Tukey's HSD comparing 60 h cell counts: P1 KO 2 vs AP1 KO 2, $p > 0.9999$ (ns); P1 KO 2 vs AP2 KO 2, $p < 0.0001$; P1 KO 2 vs AP1/AP2 KO 2, $p = 0.7951$ (ns); P2 KO 2 vs AP1 KO 2, $p < 0.0001$; P2 KO 2 vs AP2 KO 2, $p < 0.0001$; P2 KO 2 vs AP1/AP2 KO 2, $p = 0.0002$− Supplementary Fig. 7C, D). Of all the knockout clones tested, P2/AP1/ AP2 KO and P2/AP2 KO cells showed the most pronounced proliferation defects (ANOVA: $F$ (10, 22) = 104.3, $p < 0.0001$, $\eta^2 = 0.98$, 95% CI [0.97, 0.99]; Tukey's HSD comparing 60 h cell counts: Parental vs P2/ AP2 KO 1, $p < 0.0001$; Parental vs P2/AP1/AP2 KO 1, $p < 0.0001$−Fig. 4B

**Fig. 2 | Convergent evolution of PRPSAP-like homologs in Amorphea.**
**A** Structure of dimeric human PRPS1 (PDB: 2HCR), with a zoom-in of catalytic site highlighting metal binding site (Cd²⁺), AMP (represents AMP moiety of ATP), SO₄²⁻ (represents 5′-phosphate of R5P), and several conserved active site residues (magenta). D171 coordinates metal binding, K194 interacts with ATP, R196, and T225 interact with R5P, and N200 stabilizes the catalytic loop. **B** WebLogo depicting MSA of active and regulatory site residues from PRPS1 and PRPSAP2 from representative organisms in opisthokonts (*n* = 44 each). Numbers below indicate corresponding residue positions in human PRPS1 (NP_002755.1) and PRPSAP2 (NP_001340030.1). Asterisks denote residues conserved in PRPS but substituted in PRPSAP2 (also shown in **A**). **C** AlphaFold2-predicted structure of human PRPSAP2 (NP_001340030.1), with a zoom-in highlighting four non-conserved residues (magenta) at corresponding active site positions in PRPS1 shown in (**A**). AMP modeled to indicate putative ATP binding site. **D** Trimeric structure of human

PRPS1 (PDB: 2HCR). Red and magenta residues in dashed box represent dimer interface residues in bent (**B**, **C**) and parallel (**A**, **B**) dimers, respectively. Amino acid sequence of *B. subtilis* PRPS aligned with Opisthokonta PRPS homologs to identify corresponding dimer interface residues for WebLogo. Representative sequences include PRPS1 (*n* = 44), PRPSAP2 (*n* = 44) from opisthokonts, and PRPS2 (*n* = 46), PRPSAP1 (*n* = 92) from jawed vertebrates. Residue numbers based on human PRPS1 (NP_002755.1), PRPS2 (NP_002756.1), PRPSAP1 (AAH09012.1), and PRPSAP2 (NP_001340030.1). **E** PRPS paralogs from *S. cerevisiae, H. sapiens, R. potamoides, T. trahens,* and *D. rotans* showing relative NHR positions in paralogs with expanded CF and/or RF loops. Open bar for each representative species represents full poly-peptide sequences of their ancestral PRPS (bold). Amino acid positions corresponding to RF and CF loops labeled within bars. NHR insertions sites for Prs1, Prs5, PRPSAP1, PRPSAP2, PrsB, and PRPSAP-like homologs marked by triangles above/below bars. Numbers next to triangles indicate amino acid count per NHR.

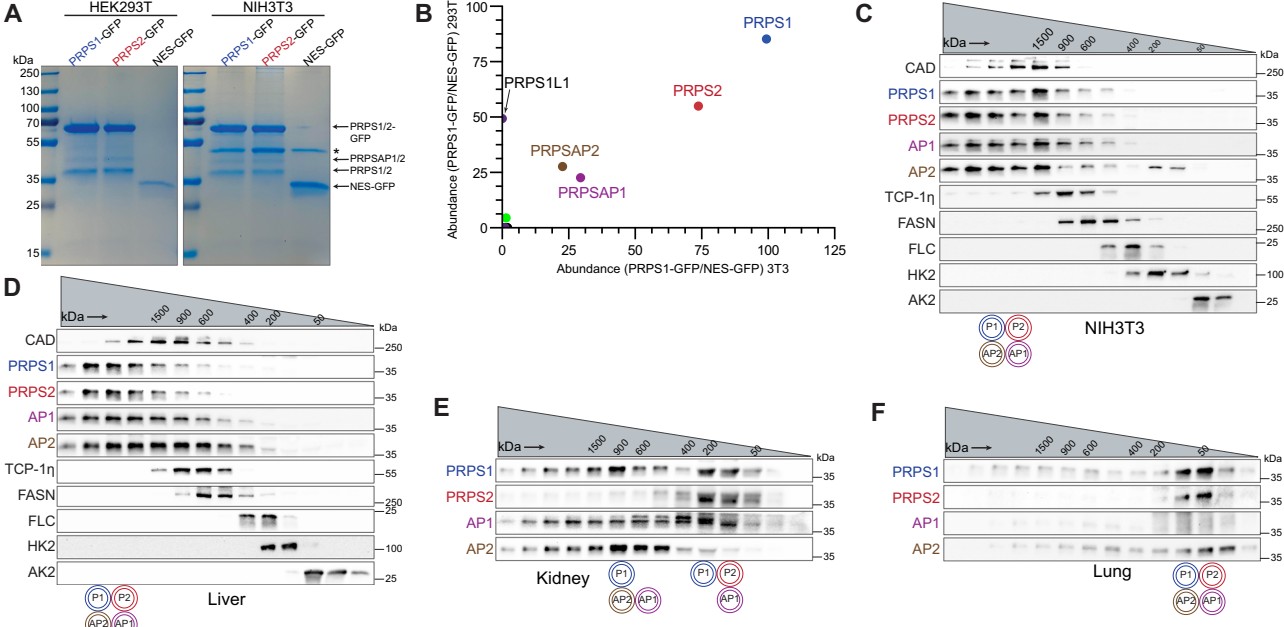

**Fig. 3 | PRPS enzymes operate as a multimeric complex that attains hetero-geneous configurations. A** SDS-PAGE followed by Coomassie stain of eluates from GFP immunoprecipitation (IP) performed in NIH3T3 and HEK293T cells stably expressing PRPS1-GFP, PRPS2-GFP, and NES-GFP. NES (Nuclear export signal)-GFP used as control. Asterisk indicates a non-specific band in eluates. **B** Scatter plot from mass spectrometry (MS) runs of eluates from GFP IP in NIH3T3 (*x*-axis) and HEK293T cells (*y*-axis) stably expressing PRPS1-GFP. Axes represent square root-transformed SEQUEST HT scores of PRPS1-GFP normalized to control. PRPS1L1 isoform detected only in HEK29T cells. **C** Western blot analysis of size exclusion chromatography (SEC) fractions collected from NIH3T3 native whole cell lysates.

Cell lysates were fractionated using a Superose 6 Increase 3.2/300 column. Immunoblots probing indicated PRPS complex members and internal standards shown. **D–F** Western blot analysis of SEC fractions from native tissue lysates of liver (**D**), kidney (**E**), and lung (**F**). Twelve-week-old male C57BL/6 mice were used. Lysates were fractionated on a Superose 6 Increase 3.2/300 column. Immunoblots probing PRPS complex members and internal standards are shown. Circular pictograms below SEC immunoblots schematize PRPS complex configurations; double circle denotes multiple copies of the protein interacting within the heteromeric complex. Coomassie staining (**A**) and western blot data (**C–F**) are representative of at least 3 biological repeats. Source data are provided as a Source Data file.

and Supplementary Fig. 7B and ANOVA: $F_{(10, 32)} = 136.0$, $p < 0.0001$, $\eta^2 = 0.98$, 95% CI [0.98, 1.00]; Tukey's HSD comparing 60 h cell counts: Parental vs P2/AP2 KO 2, $p < 0.0001$; Parental vs P2/AP1/AP2 KO 2, $p < 0.0001$—Supplementary Fig. 7C, D). Collectively, these findings demonstrate the importance of partnerships within the PRPS enzyme complex in establishing specific configurations that are vital for maintaining optimal PRPS activity in mammalian cells.

To test whether an increased anabolic stimulus would circumvent or augment the diminished PRPS-controlled proliferative capacity, we overexpressed mutant oncogenic Ras, which is known to transform cells in part by rewiring metabolism to increase pentose phosphate pathway flux and nucleotide production[45]. Upon transduction of NIH3T3 parental cells and P2/AP1/AP2 KO cells with H-Ras^G12V, we observed that parental cells readily formed colonies in soft agar medium, while P2/AP1/AP2 KO

cells produced significantly fewer and smaller colonies suggesting PRPS activity may impose a strict ceiling on metabolic flux (ANOVA: $F_{(2, 6)} = 69.62$, $p < 0.0001$, $\eta^2 = 0.96$, 95% CI [0.91, 1.00]; Tukey's HSD: Parental vs P2/AP1/AP2 KO 1, $p = 0.0011$; Parental vs P2/AP1/AP2 KO 2, $p < 0.0001$—Fig. 4C–E). To understand the basis for impaired cell proliferation, we first checked cell cycle profiles which revealed there was no cell cycle arrest in the knockout clones (Supplementary Fig. 7E). Additionally, cleaved-PARP1 levels were comparable across parental and knockout cells, indicating that apoptosis is not induced in the knockout cells (Supplementary Fig. 7F). Taken together, these results point toward a slower progression through the cell cycle in knockout cells likely due to decreased PRPP synthesis rates.

To pinpoint the metabolic basis for decreased proliferative capacity of P2/AP1/AP2 KO cells, we first measured total ATP levels in

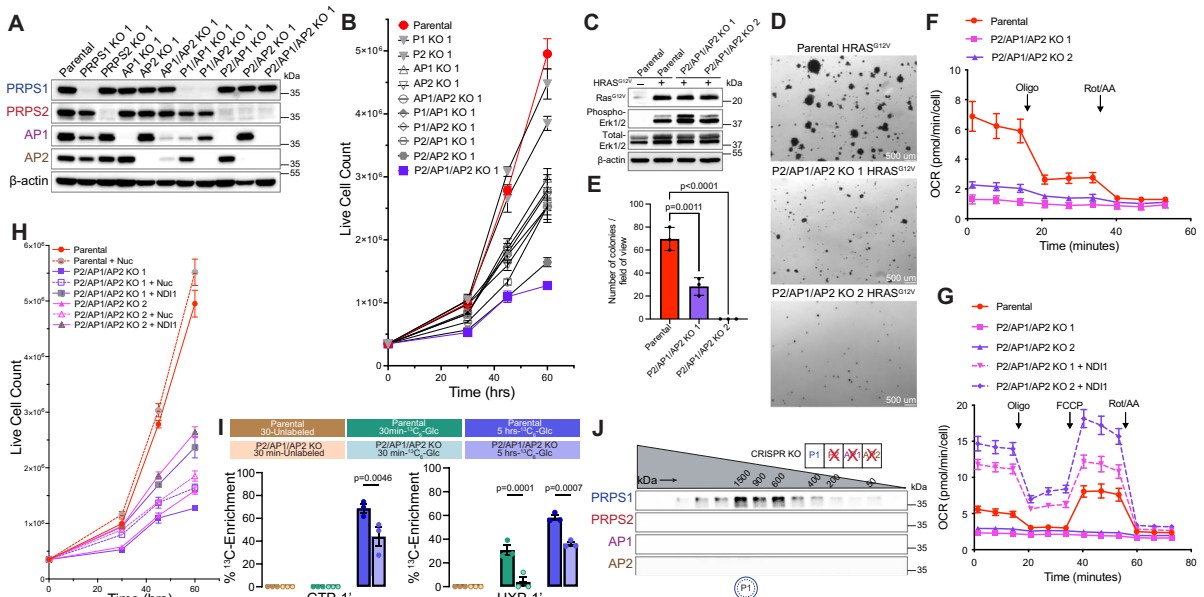

**Fig. 4 | Cells exclusively expressing PRPS1 display metabolic defects and decreased cellular proliferation. A** Western blot validating CRISPR-Cas9-generated isogenic knockout cell lines. **B** Proliferation of NIH3T3 parental and knockout cell lines generated in (**A**) ($n = 3$ technical replicates). **C** Western blot validating HRAS$^{G12V}$-overexpression in NIH3T3 parental and P2/AP1/AP2 KO cell lines. Phospho-MAPK (Erk1/2) (T202/Y204) used as a marker for activation of signaling pathways upon HRAS$^{G12V}$ overexpression. **D** Representative images from soft agar colony formation assay performed in NIH3T3 parental and P2/AP1/AP2 KO cells expressing HRAS$^{G12V}$. Scale bar, 500 μm. **E** Quantification of colonies from (**D**) ($n = 3$ experimental replicates). **F** Oxygen consumption rate (OCR) measured by Seahorse ATP Rate Assay in NIH3T3 parental and P2/AP1/AP2 KO cell lines ($n = 8$ technical replicates). **G** OCR measured by Seahorse mitochondrial stress tests in NIH3T3 parental and NDI1-expressing P2/AP1/AP2 KO cell lines ($n = 7$ technical

replicates). **H** Proliferation of nucleoside supplemented and NDI1 expressing NIH3T3 P2/AP1/AP2 cell lines ($n = 3$ technical replicates). **I** $^{13}C_6$-glucose metabolic labeling performed in NIH3T3 parental and P2/AP1/AP2 KO cell lines for 30 min and 5 h. Unlabeled, $^{13}$C-labeled (30 min), and $^{13}$C-labeled (5 h) samples are represented in brown, green, and blue colors, respectively. $^{13}$C-enrichments quantified from $^{1}$H-NMR spectra ($n = 3$ experimental replicates). **J** Western blot analysis of SEC fractions collected from NIH3T3 P2/AP1/AP2 KO native whole-cell lysates. Cell lysates were fractionated on a Superose 6 Increase 3.2/300 column. In the pictogram below SEC immunoblots, the double circle with dotted inner circle denotes multiple copies of PRPS1 forming homo-oligomers. Data are represented as mean ± SD for (**B**, **E–H**) and mean ± SEM for (**I**). Statistical comparisons made using one way ANOVA followed by Tukey's HSD post hoc test (**E**, **I**). Source data are provided as a Source Data file.

---

parental and P2/AP1/AP2 KO cells and observed a slight decrease in P2/AP1/AP2 KO cells (ANOVA: F (2, 6) = 12.63, $p = 0.0071$, $\eta^2 = 0.81$, 95% CI [0.57, 0.99]; Tukey's HSD: Parental vs P2/AP1/AP2 KO 1, $p = 0.0058$; Parental vs P2/AP1/AP2 KO 2, $p = 0.0666$–Supplementary Fig. 7G). However, that modest difference was not sufficient to trigger energy stress as phosphorylation of AMPK remained consistent between parental and knockout clones (Supplementary Fig. 7F). Using the Seahorse assay to evaluate mitochondrial and glycolytic energy production, we found that, despite comparable total ATP production rates, P2/AP1/AP2 KO cells had significantly lower respiration-linked ATP production rates, which were compensated by an increase in glycolytic ATP production rates compared to wild-type cells (For glyco ATP–ANOVA: F (2, 21) = 23.67, $p < 0.0001$, $\eta^2 = 0.69$, 95% CI [0.52, 0.87]; Tukey's HSD: Parental vs P2/AP1/AP2 KO 1, $p < 0.0001$; Parental vs P2/AP1/AP2 KO 2, $p = 0.0315$ and for mitoATP–ANOVA: F (2, 21) = 223.5, $p < 0.0001$, $\eta^2 = 0.96$, 95% CI [0.92, 0.99]; Tukey's HSD: Parental vs P2/AP1/AP2 KO 1, $p < 0.0001$; Parental vs P2/AP1/AP2 KO 2, $p < 0.0001$–Supplementary Fig. 7H). Measuring mitochondrial oxygen consumption rates (OCR) revealed a striking loss of mitochondrial respiration in P2/AP1/AP2 KO cells, perhaps reflective of altered redox homeostasis[46,47] (ANOVA: F (2, 21) = 198.6, $p < 0.0001$, $\eta^2 = 0.94$, 95% CI [0.92, 0.98]; Tukey's HSD: Parental vs P2/AP1/AP2 KO 1, $p < 0.0001$; Parental vs P2/AP1/AP2 KO 2, $p < 0.0001$–Fig. 4F and Supplementary Fig. 7I). To rescue this mitochondrial defect, we overexpressed NDI1 (yeast complex I)[48] in P2/AP1/AP2 KO cells. NDI1 successfully increased mitochondrial OCR, including both basal and maximal rates, and enhanced ATP production (For Basal OCR–ANOVA: F (4, 30) = 888.9, $p < 0.0001$, $\eta^2 = 0.99$, 95% CI [0.99, 1.00]; Tukey's HSD: Parental vs P2/AP1/AP2 KO 1, $p < 0.0001$; Parental vs P2/AP1/AP2 KO 2, $p < 0.0001$; P2/

AP1/AP2 KO 1 vs P2/AP1/AP2 KO 1 + NDI1, $p < 0.0001$; P2/AP1/AP2 KO 2 vs P2/AP1/AP2 KO 2 + NDI1, $p < 0.0001$ and for Maximal OCR–ANOVA: F (4, 30) = 533.2, $p < 0.0001$, $\eta^2 = 0.99$, 95% CI [0.98, 1.00]; Tukey's HSD: Parental vs P2/AP1/AP2 KO 1, $p < 0.0001$; Parental vs P2/AP1/AP2 KO 2, $p < 0.0001$; P2/AP1/AP2 KO 1 vs P2/AP1/AP2 KO 1 + NDI1, $p < 0.0001$; P2/AP1/AP2 KO 2 vs P2/AP1/AP2 KO 2 + NDI1, $p < 0.0001$ and For ATP OCR–ANOVA: F (4, 30) = 872.0, $p < 0.0001$, $\eta^2 = 0.99$, 95% CI [0.99, 1.00]; Tukey's HSD: Parental vs P2/AP1/AP2 KO 1, $p < 0.0001$; Parental vs P2/AP1/AP2 KO 2, $p < 0.0001$; P2/AP1/AP2 KO 1 vs P2/AP1/AP2 KO 1 + NDI1, $p < 0.0001$; P2/AP1/AP2 KO 2 vs P2/AP1/AP2 KO 2 + NDI1, $p < 0.0001$–Fig. 4G and Supplementary Fig. 7J). Improving mitochondrial respiration offered slightly better rescue than nucleoside supplementation alone, however, neither experimental manipulation, alone or in combination, was sufficient to fully restore proliferation rates to that of parental cells suggesting that P2/AP1/AP2 KO cells suffer from a more widespread metabolic defect (Fig. 4H). To investigate whether a decreased flux in PRPP-consuming metabolic routes might be driving these metabolic defects including mitochondrial dysfunction in P2/AP1/AP2 KO cells, we performed $^{13}C_6$-glucose tracing experiments, which revealed that P2/AP1/AP2 KO cells indeed have decreased pentose phosphate pathway flux and nucleotide production in addition to changes in glycolysis, creatine phosphate pathway, and choline metabolism (For GTP-1'– ANOVA: F (5, 12) = 67.30, $p < 0.0001$, $\eta^2 = 0.97$, 95% CI [0.91, 1.00]; Tukey's HSD: Parental (5 h-$^{13}C_6$ Glu) vs P2/AP1/AP2 KO (5hr-$^{13}C_6$Glu), $p = 0.0046$ and for UXP-1'– ANOVA: F (5, 12) = 83.35, $p < 0.0001$, $\eta^2 = 0.97$, 95% CI [0.94, 1.00]; Tukey's HSD: Parental (30 min-$^{13}C_6$Glu) vs P2/AP1/AP2 KO (30 min-$^{13}C_6$Glu), $p = 0.0001$; Parental (5 h-$^{13}C_6$Glu) vs P2/AP1/AP2 KO (5hr-$^{13}C_6$Glu), $p = 0.0007$–Fig. 4I and Supplementary Data 7). These

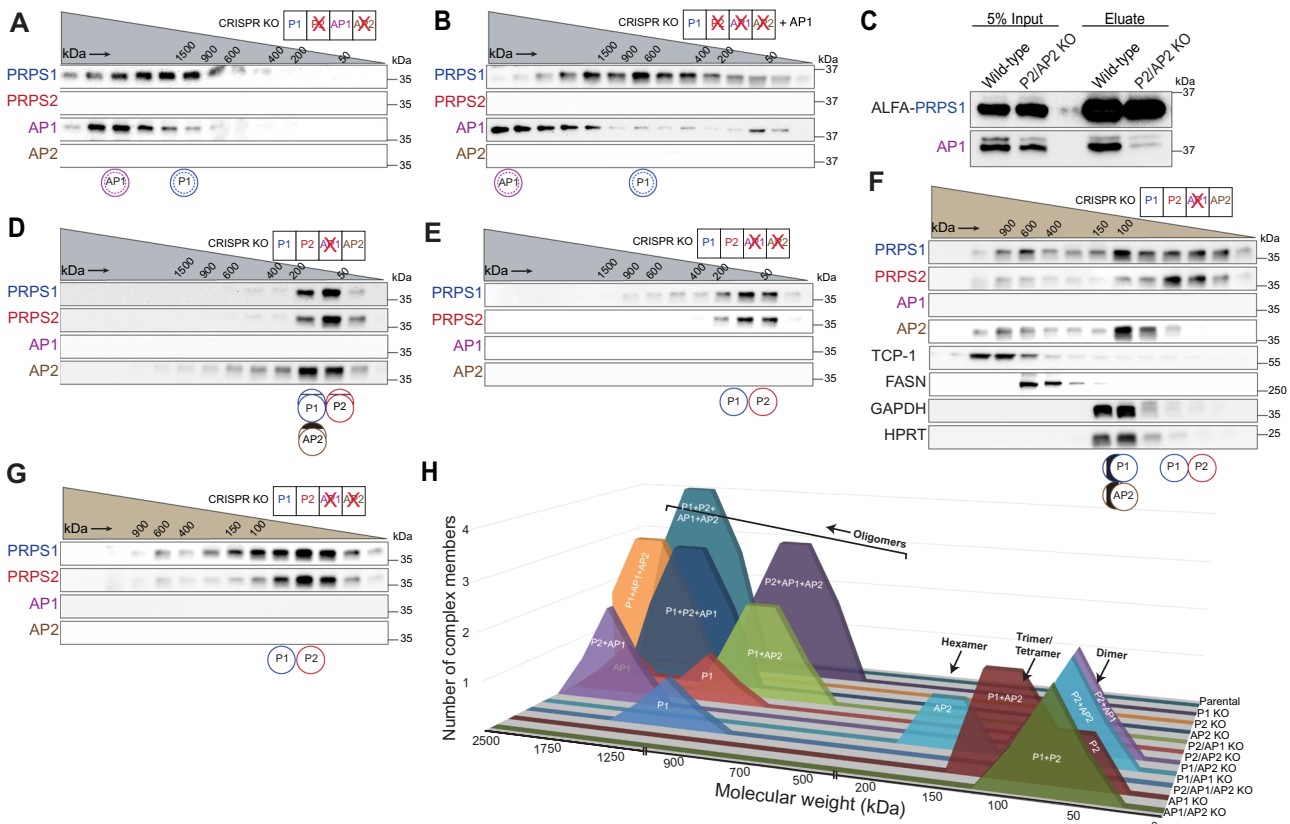

**Fig. 5 | PRPSAPs are molecular scaffolds with preferential binding partners among PRPS isozymes. A, B** Western blot analysis of SEC fractions collected from native whole cell lysates of NIH3T3 P2/AP2 KO cells (**A**) and P2/AP1/AP2 KO cells stably expressing AP1 (**B**). **C** ALFA pulldown from whole cell extracts of NIH3T3 parental and P2/AP2 KO cells transiently transfected with PRPS1-ALFA. **D–G** Western blot analysis of SEC fractions collected from native whole-cell lysates of NIH3T3 AP1 KO cells (**D**), AP1/AP2 KO cells (**E**), AP1 KO cells (**F**), and AP1/AP2 KO cells (**G**). **H** Summary comparing SEC profiles from NIH3T3 parental and CRISPR KO lines. The x-axis represents relative molecular weight of the complex, and the y-axis represents the number of PRPS complex members interacting in the complex. Cell lysates for (**A, B, D,** and **E**) were fractionated on a Superose 6 Increase 3.2/300 column, while cell lysates for (**F**) and (**G**) were fractionated on a Yarra SEC-2000 column. Circular pictograms below SEC immunoblots schematize PRPS complex configurations. Double circle denotes multiple copies of the protein interacting in a heteromeric complex. Double circle with dotted inner circle denotes multiple copies forming homo-oligomers. Single circle denotes a single protein interacting within the complex. Circle with inner vertical lines denotes proteins forming a trimer or tetramer. Western blot data (**A–G**) are representative of at least 2 biological repeats. Source data are provided as a Source Data file.

results demonstrate that PRPS1 as a standalone enzyme in cells is insufficient to sustain the metabolic flux necessary to keep up with the demands of proliferation.

Given that wild-type cells produce high molecular weight PRPS complex assemblies, we hypothesized that PRPS1 alone might be insufficient to form the large oligomers required for optimal activity. However, we observed that PRPS1 alone could still form higher-order homotypic assemblies (Fig. 4J) consistent with recent cryo-EM structures showing filamentous assembly of PRPS1[35,41]. Since AP1/AP2 KO cells proliferate faster than P2/AP1/AP2 KO cells (ANOVA: $F_{(10, 22)} = 104.3$, $p < 0.0001$, $\eta^2 = 0.98$, 95% CI [0.97, 0.99]; Tukey's HSD comparing 60 h cell counts: AP1/AP2 KO 1 vs P2/AP1/AP2 KO 1, $p < 0.0001$–Fig. 4B and Supplementary Fig. 7B and ANOVA: $F_{(10, 32)} = 136.0$, $p < 0.0001$, $\eta^2 = 0.98$, 95% CI [0.98, 1.00]; Tukey's HSD comparing 60 h cell counts: AP1/AP2 KO 2 vs P2/AP1/AP2 KO 2, $p < 0.0001$–Supplementary Fig. 7C, D), we speculated that in AP1/AP2 KO cells, PRPS1 may assemble with PRPS2 to form mixed filaments thereby enhancing the complex activity, especially since PRPS2 has been shown to form filaments in vitro as well[36]. Paradoxically, reexpressing exogenous PRPS2 in PRPS1-only (P2/AP1/AP2 KO) cells restricted PRPS1 into smaller complexes with PRPS2 indicative of an improved activity over PRPS1 homo-oligomers (Supplementary Fig. 7K), which suggests that filamentation is not a strict correlate of PRPS activity. Altogether, our findings suggest that homotypic PRPS1 assemblies, which can be disrupted by PRPS2, are aberrant and sub-optimal configurations for cells.

**Organizing principles of PRPS complex assembly**
Based on our findings that homo-oligomerization of PRPS1 is detrimental to cellular metabolism, we hypothesized that P2/AP2 KO cells, which share similar proliferative defects with P2/AP1/AP2 KO cells (Fig. 4B and Supplementary Fig. 7B), might contain such aberrant homotypic assemblies. Indeed, in P2/AP2 KO cells, PRPS1 and AP1 did not bind to each other; instead, they formed homo-oligomers of distinct sizes (Fig. 5A), a result that was recapitulated by overexpressing AP1 in PRPS1-only cells (Fig. 5B). We further confirmed these results by performing PRPS1 IP in P2/AP2 KO cells where we observed minimal interactions between PRPS1 and AP1 (Fig. 5C). Surprisingly, these results demonstrate that, despite the strict conservation of parallel and bent dimer interfaces between PRPS isozymes and associated proteins, there likely exists a preferential selection of binding partners within the complex.

Building on the concept of specific partnerships, we next investigated the preferential binding modalities within the PRPS complex. In P2/AP1 KO cells, PRPS1 and AP2 together formed higher-order structures (Supplementary Fig. 8A) resembling kidney—which exhibited low AP1 expression—where PRPS1 and AP2 were found in the same ~900 kDa MW range (Fig. 3E). Next, we asked whether PRPS2 and AP1

can similarly interact with each other. Indeed, in P1/AP2 KO cells, PRPS2 and AP1 formed higher-order heterotypic assemblies (Supplementary Fig. 8B). These results suggest an ordered assembly of the complex dictated by preferential heteromeric binding between PRPS1 and AP2—partners that have co-evolved since the early opisthokonts—as well as between PRPS2 and AP1, which originated together in the ancestor of jawed vertebrates thereby establishing asymmetric heteropairing as the primary binding mode.

To identify the primary determinant of formation of competent HMW PRPS complex, we characterized complex assembly potential in remaining isogenic knockout series. We found that the loss of PRPS1, PRPS2 or AP2 alone did not substantially alter the formation of higher order heterotypic assemblies (Supplementary Fig. 8C–E). Interestingly, in the absence of AP1—observed in both AP1 KO and AP1/AP2 KO cells – we detected a markedly smaller PRPS complex, indicating a major role for AP1 in governing HMW complex formation (Fig. 5D, E). In line with this, we noted that in P1/AP1 KO cells, PRPS2 and AP2 were unable to assemble into a HMW complex (Supplementary Fig. 8F).

To investigate whether AP1 KO and AP1/AP2 KO cells form distinct PRPS complex configurations, we employed a Yarra SEC-2000 column (optimal resolution range −1 kDa to 300 kDa, better for smaller MW) as Superose 6 Increase 3.2/300 column (optimal resolution range −5 kDa to 5 MDa) produced identical elution profiles for both knockout lines. In AP1 KO cells, PRPS1 preferentially binds with AP2 to form heterotrimers or heterotetramers, while PRPS2 primarily forms dimers (Fig. 5F). In AP1/AP2 KO cells, we predominantly observe PRPS1-PRPS2 heterodimers (Fig. 5G) reminiscent of lung, which exhibited low AP1 and AP2 expression (Fig. 3F). This finding aligns with our previous results showing that reintroducing PRPS2 to PRPS1-only cells restricts PRPS1 homo-oligomerization (compare Fig. 5G with Supplementary Fig. 7K). Here, we demonstrate that AP2 outcompetes PRPS2 for binding to PRPS1, likely because the PRPS1·PRPS2 complex must form a bent dimer for catalysis, preventing the assembly of a parallel dimer required for allosteric site formation, which renders it insensitive to feedback regulation. In contrast, the PRPS1-AP2 complex, capable of forming either a heterotrimer or heterotetramer, can assemble both parallel and bent dimers, thereby remaining responsive to allosteric regulation. Altogether, we demonstrate preferential binding between PRPS1-AP2 and PRPS2-AP1, establish dimer pairing rules (Supplementary Fig. 8G), and attribute stimulation of PRPS complex elongation as an emergent property of AP1 (summary in Fig. 5H).

### Translational control of PRPS enzyme assembly

Because disordered NHRs are prominent features of PRPSAPs, and intrinsically disordered regions are known to enable biomolecular condensate formation[49], we hypothesized that the NHR unique to PRPSAP1 might promote PRPS complex elongation. To test this, we overexpressed an NHR-deletion construct of AP1 in AP1 KO cells, which fully restored the formation of higher order heterotypic PRPS complex (Fig. 6A). These results indicate that NHRs, a feature shared between AP1 and AP2 isoforms, do not play a direct role in mediating complex assembly. However, the positive selection of NHRs in PRPSAPs suggests that they may influence the flexibility or regulation of the complex in other ways.

To identify the structural elements that distinguish AP1 from AP2 and confer AP1 its complex elongation properties, we generated four chimeric constructs by swapping regions between AP1 and AP2 and overexpressed them individually in AP1/AP2 KO cells (Supplementary Fig. 9A). Only the chimera with N-terminus of AP2 (residues 1–95) failed to restore the formation of higher order heterotypic PRPS complex, whereas the other three constructs successfully formed HMW PRPS complex, confirming the N-terminus of AP1 as the distinguishing feature that promotes HMW complex assembly (Fig. 6B and Supplementary Fig. 9B–D). As expected, replacing AP2's N-terminus with the corresponding residues from AP1 promoted the formation of larger

multimers (Fig. 6C). Notably, excluding the flexible residues within the NHRs, the N-terminus (residues 1–95) accounts for over 50% of the amino acid differences between human AP1 and AP2 (Supplementary Fig. 9E). A comparative analysis of the N-termini across jawed vertebrates revealed that these differences have been maintained since the divergence of AP1 from AP2 (Supplementary Fig. 9F). This highlights how rapid adaptive changes at the N-terminus of AP1 facilitate complex elongation, possibly reflecting an evolutionary response to the co-emergence of PRPS2 that restricts PRPS1 elongation.

While analyzing the N-terminal residues of AP1 and AP2, we noticed that the transcripts of AP1 and AP2 include an upstream alternative translation start site (TSS), which encodes an additional 29 and 12 amino acids at the N-terminus for murine AP1 and AP2, respectively (Fig. 6D, E). These upstream sequences with alternative start sites for AP1 and AP2 have been positively selected for since their emergence in Osteichthyes and Amniota, respectively (Fig. 6F and Supplementary Fig. 9G, H). To understand the influence of long and short isoforms, we introduced them individually into AP1/AP2 KO cells and performed SEC. Interestingly, exogenous expression of the short AP1 isoform completely restored the complex size whereas the longer AP1 isoform failed to interact with and incorporate PRPS1 and PRPS2 into higher-order structures (Fig. 6G, H). In contrast, both long and short isoforms of AP2 failed to restore the complex size (Fig. 6I, J). These results conclusively identify the short AP1 isoform as the primary driver of PRPS complex elongation and highlight the regulated assembly of PRPS complex via translational control, mediated by the inclusion or exclusion of N-terminal leader sequences.

## Discussion

Eukaryogenesis involved expansion in cell volume, genome size, number of protein-coding genes and regulation of gene expression, along with the addition of metabolically demanding compartmentalized machineries, all fueled by an energy boost from mitochondria[50,51]. The complexity of eukaryotes included the diversification of metabolic enzymes, likely providing the LECA with an adaptive advantage to inhabit various ecological niches and increasing the likelihood of survival for diverse lineages. Here, we showed that acquisition and maintenance of multiple PRPS homologs is a key trait of the eukaryotic metabolic system, underscored by our demonstration that PRPS1 homo-oligomers do not function effectively to sustain eukaryotic metabolism. It is clear from our eukaryote-wide survey of PRPS homologs that later symbiotic events, genetic reshuffling, and gene duplications have provided fertile ground for sculpting a PRPS armament enriched with evolutionary innovations to enable species to adapt their metabolism. Our case study on the origins of the mammalian PRPS complex demonstrate how a single PRPS gene, over a vast evolutionary timescale, ultimately gives rise to multiple discrete homologs while maintaining intermolecular interactions to create a tunable multi-component metabolic assembly (Fig. 7A)[52].

The precise positive and negative selection pressures that shaped PRPS homologs after their emergence are hard to determine post hoc, but some functional changes may offer insights. Despite losing key active site residues necessary for independent PRPP generation, the conserved FLAG region (Fig. 2B) in PRPSAPs and PRPSAP-like orthologs suggests they might still form a minimal functional unit—a dimer—with a PRPS isozyme. Interestingly, we have shown that independent gene duplications throughout evolution have enabled remodeling of flexible loops, as seen in the expansion of regulatory loops (in Prs5 orthologs) and CF loops (in PRPSAP2/PrsB/PRPSAP-like orthologs), and positive selection has preserved such innovations. An analysis of more than 30,000 proteins has shown that nature relies on a limited repertoire of basic structures (domains, motifs, and folds) to perform a wide range of functions[53]. Enzymes achieve functional innovations by altering the flexible loop structures exposed on their surfaces[54,55], including those in PRPP-utilizing enzymes[56]. In the PRPS complex,

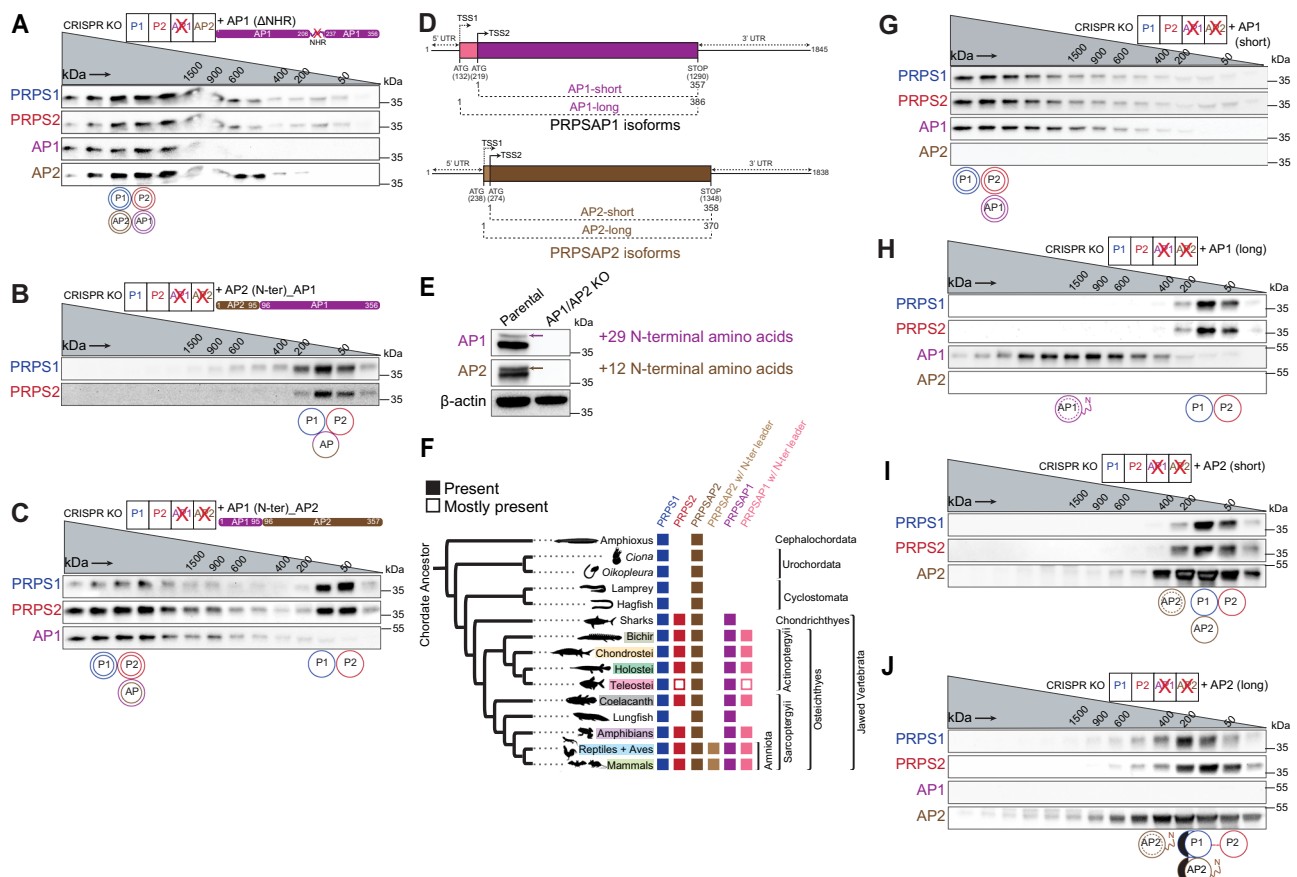

**Fig. 6 | Translation control of PRPS complex assembly. A–C** Western blot analysis of SEC fractions collected from native whole-cell lysates of NIH3T3 AP1 KO cells stably expressing AP1 lacking the non-homologous region (NHR) (**A**), AP1/AP2 KO cells stably expressing chimeric AP1 with AP2's N-terminus (residues 1–95) **B** and chimeric AP2 with AP1's N-terminus (residues 1–95) (**C**). **D** Schematic representation of alternative start sites in mammalian AP1 and AP2 and their consequent translation into short and long isoforms. Base positions correspond to the mouse homologs. TSS1 and TSS2 represent upstream and downstream translation start sites, respectively. **E** Multiple isoforms for AP1 and AP2 detected under optimal SDS-PAGE resolution. Arrows in immunoblots indicate the longer isoforms of AP1 and AP2, corresponding to N-terminal leader sequences of 29 and 12 amino acids, respectively. **F** Phylogenetic distribution of PRPS homologs (PRPS1, PRPS2, PRPSAP2, PRPSAP2 with N-terminal leader sequence, PRPSAP1, and PRPSAP1 with N-terminal leader sequence) in chordates. Presence/absence noted across the tree. PRPSAP2 and PRPSAP1 isoforms with N-terminal leader sequences emerged in ancestors of Amniota and Osteichthyes, respectively. **G–J** Western blot analysis of SEC fractions collected from NIH3T3 AP1/AP2 KO cells stably expressing the short isoform of AP1 (**G**), long isoform of AP1 (**H**), short isoform of AP2 (**I**), and long isoform of AP2 (**J**). Cell lysates were fractionated on a Superose 6 Increase 3.2/300 column. Circular pictograms below SEC immunoblots schematize PRPS complex configurations. Double circle denotes multiple copies of the protein interacting in a heteromeric complex. Double circle with dotted inner circle denotes multiple copies forming homo-oligomers. Single circle denotes a single protein interacting within the complex. Circle with inner vertical lines denotes proteins forming a trimer or tetramer. Western blot data (**A–C**, **E**, **G–J**) are representative of at least 2 biological repeats. Source data are provided as a Source Data file.

these disordered CF loops could be primed by allosteric mechanisms and post-translational modifications (PTMs), allowing them to act as dynamic switches that can rapidly adapt to the cell's metabolic needs by sampling various conformational states. For instance, the movement of the CF loop in PRPSAP1/2 could influence the kinetics of catalytic loop opening in adjacent PRPS1/2[35]. Different loop conformational substates might influence access to allosteric and PTM sites in the neighboring PRPS isozymes[57–59], as well as modulate intra and intermolecular interactions mediated via allostery[5]. Additionally, PRPSAPs exhibit poor conservation in the RF loop residues that are essential for formation of allosteric sites I and II (Subunit B in Supplementary Fig. 10A–D)[6,8]. The extent of PRPSAP-mediated regulation likely depends on the specific subunit arrangement in a PRPS-PRPSAP dimer/trimer and even across protomers of a filament, as filament interfaces are highly conserved in PRPSAPs, similar to PRPS isozymes[35,36]. Interestingly, our evolutionary analyses and biochemical data also reveal that the PRPS complex predominantly assembles as heterogenous asymmetric heteromers, formed from minimal functional units that consist of an isozyme and an associated protein or two

different isozymes. However, the structural basis for these heteromeric partnerships remain unclear, as all published studies to date are limited to homomeric configurations. Our findings pave the way for future structural and biochemical studies aimed at deciphering the molecular underpinnings of different heterogenous PRPS complex configurations.

Our phylogenomic analysis uncovered several instances in PRPS evolution where the loss of Class II was followed by expansion of Class I PRPS homologs. These newer Class I homologs were enriched with novel functional and regulatory features that may compensate for loss Class II PRPS, which lacks allosteric inhibition by downstream nucleotides. Indeed, unlike PRPS1, PRPS2 is highly feedback-resistant[60], and we have shown that it also suppresses formation of supramolecular assemblies with PRPS1, adding another dimension to PRPS2's ability to stimulate PRPP production. We propose that AP1 concurrently emerged to counteract PRPS2 by preferentially binding to it and restoring formation of higher-order structures, reminiscent of PRPS1's abilities. Additionally, the duplication of AP1 from AP2 led to significant structural changes, particularly at the N-terminus, enabling

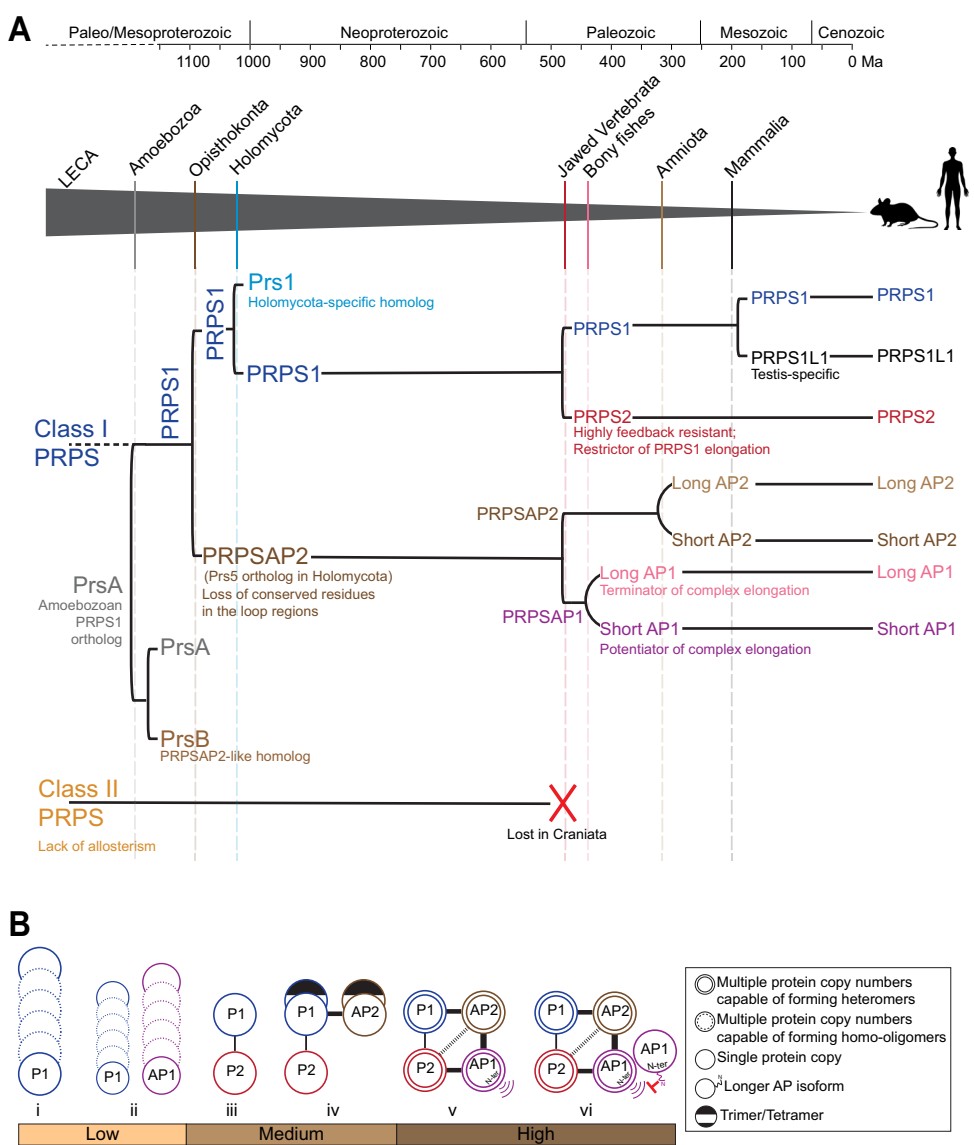

**Fig. 7 | Timeline of PRPS complex evolution in mammals and implications of distinct PRPS assembly states on cellular fitness. A** Chronological duplication events of a Class I PRPS encoding gene in indicated eukaryotes. Grey triangle traces the evolutionary trajectory leading to the emergence of mammalian PRPS homologs. Solid vertical lines within the triangle denote the origin of major lineages; dashed lines indicate significant duplication events giving rise to PRPS homologs, with notable functional innovations occurring post-duplication. Class II PRPS, likely present in the LECA, is lost in craniates, followed by rapid expansion of Class I PRPS homologs in jawed vertebrates. Left square bracket indicates a gene duplication event; left round bracket indicates an extension at the N-terminus. Molecular clock-based evolutionary timeline (in millions of years before present, Ma) adapted from ref. 52. **B** Association of different PRPS complex configurations with cellular fitness in mammals (i) PRPS1 alone undergoes self-assembly to form homo-oligomers. (ii) PRPS1 and AP1 cannot interact directly and form separate homo-oligomers. (iii) PRPS2 binds PRPS1 to form a dimer and disrupts PRPS1 homotypic assemblies. (iv) AP2 preferentially binds with PRPS1 to form a trimer/tetramer to nucleate the complex. (v) AP1 preferentially binds with AP2 and PRPS2 to elongate the PRPS complex via its N-terminus. (vi) The longer isoform of AP1 with flexible N-terminal extension likely caps the complex. The weight of the connectors between the circles representing proteins reflects the predicted strength of protein-protein interactions; heavier lines indicate stronger predicted interactions.

it to drive higher-order heterotypic assemblies. The N-terminus is crucial, as evolution has introduced N-terminal leader sequences in both AP1 and AP2, further diversifying their roles. Similar to Myc-driven translational regulation of PRPS2[61], translation control emerges as a crucial driver of PRPS complex assembly via TSS selection in PRPSAPs providing an important link between major anabolic sinks and metabolic flux determining enzymes, however, the mechanisms governing TSS selection and the physiological settings under which this regulation is important remains unknown.

Mammalian metabolic complexes are known to exhibit various molecular forms to tailor enzymatic configurations according to

metabolic demands[62–65]. Our discovery that the PRPS complex exhibits altered architecture across different tissues, combined with recent evidence of varied purine synthesis rates across tissues[66] where PRPS is a major determinant of purine production[67–69], suggests that the flexible nature of the PRPS complex is vital for metabolic adaptation. The quality and magnitude of effect that PRPS complex modulation has on cellular biochemistry may be cell-type specific[70]. Our genetic knockout studies in cells support the existence of multiple configurations of PRPS complex, which can result in varying levels of cellular fitness (Fig. 7B). These myriad assembly states, involving changes in stoichiometry, topology, dynamic conformations, and differential affinities, have

significant implications for regulatory and catalytic properties. Our research lays the groundwork for future studies focused on understanding the mechanisms underlying the upstream signals that regulate PRPS complex assembly and the associated activity and metabolic output of particular PRPS complex configurations. Ultimately, this study advocates for a paradigm shift, highlighting that PRPS activity is regulated at the level of the complex—shaped by over a billion years of evolution—which may explain the variable penetrance and pleiotropic effects seen in family members with the same PRPS disease-causing mutations.

## Methods

### Database search and annotation for sequences

Amino acid sequences from well annotated PRPS enzymes from model organisms were used as templates to identify homologs in the NCBI database using BLAST[71]. Hits were subsequently curated manually based on significant matches, protein domains, and conserved motifs.

For manual annotation, datasets were retrieved from Sequence Read Archive[72]. Selection criteria prioritized datasets with high-quality sequencing reads and comprehensive metadata, ensuring suitability for downstream protein annotation. Quality control checks were performed using BlastP versus eukaryotic or prokaryotic domains. Open reading frames were identified within transcripts, and potential protein sequences were deduced based on canonical start and stop codons, as well as sequence homology to well-annotated PRPS proteins.

For splicing analysis, genomic DNA or mRNA sequences used to determine splice site junctions for each PRPS homolog were obtained from the NCBI/Ensembl[73] databases. When exonic and intronic sequences were not explicitly provided, we used homologous sequences to predict amino acids and applied splice junction rules (canonical GT/AG and non-canonical GC/AG) to define exon-intron boundaries. The corresponding exonic and intronic sequences for each identified homolog used in the analysis are available on Figshare. Where available, splice prediction was corroborated with transcriptomics of the same species or nearest available relative.

For synteny analysis, PRPS1, PRPS2, PRPSAP2, and PRPSAP1 encoding genes were mapped onto the genomes of various vertebrates using the NCBI and Ensembl genome browsers. The protein accession numbers for each mapped PRPS homolog are available on figshare. Unannotated neighboring genes are presented as uncharacterized genes and labeled with their GeneID.

Multiple sequence alignment (MSA) was conducted using Clustal Omega[74] or MUSCLE[75]. A sequence logo representing desired regions was generated using the WebLogo[76] tool online, based on the MSA results. Multiple logos were aligned using MetaLogo[77].

### PRPS1 structure generation

The PRPS1 crystal structure with the ID−2hcr was extracted from the PDB. Missing residues in the N-terminus, C-terminus, and CF regions were modeled using MODELLER[78] in UCSF ChimeraX[79]. Multiple loop models were generated, and the final selections were based on the lowest normalized discrete optimized protein energy score (zDOPE), where a negative score denotes better predictions.

### AlphaFold2 structure predictions

AlphaFold modelling[80] was performed using the Alphafold2_mmseq2[81] notebook, which was run on Google Collaboratory cloud[82]. AlphaFold's top-scoring models were ranked from 1 to 5 by per-residue Local Distance Difference Test (pLDDT) scores (a per-residue estimate of the prediction confidence on a scale from 0 to 100). The model with the highest average pLDDT scores was used to assess the predicted structures. The selected model was visualized and analyzed using ChimeraX.

### Cell growth assay

Cells were seeded onto a 6 cm plate at a density of 350,000 cells/plate. Cells counts were counted at three time points: 30, 45, and 60 h after seeding. Cell counting was performed using trypan blue staining and a hemocytometer.

### Plasmids and transfection

*Prps1* cDNA (Horizon Discovery #MMM1013-202859297), *Prps2* cDNA (previously described[61]), and *PRPSAP1* cDNA (Horizon Discovery #MHS6278-202757585) were subcloned into the pMSCV-puro vector, which harbors GFP at the C-terminus, using the SalI and EcoRI sites. *PRPSAP2* cDNA (Horizon Discovery #MHS6278-202759946) was subcloned into the pMSCV vector with GFP at the C-terminus using Gibson assembly. *PRPSAP1* and *PRPSAP2* cDNA from Horizon Discovery encode the short PRPSAP1 isoform and the long PRPSAP2 isoform, respectively. Construct encoding yeast *NDI1* retroviruses (Addgene #72876) has been previously described[83]. *Myc* cDNA in pWZL Blast myc (Addgene #10674) was replaced with *H-Ras V12* cDNA from pBABE-puro H-Ras V12 (Addgene #9051) to generate pWZL H-Ras V12 blast.

Lentiviral expression plasmids for PRPS complex members were derived as follows. First, the blasticidin expression cassette from pWZL Blast myc (Addgene #10674) was subcloned into FUGW_bleo (Addgene # 14883) to replace bleomycin. All four cDNAs encoding PRPS1, PRPS2, PRPSAP1, and PRPSAP2 were PCR amplified from pMSCV-puro vector and ligated with PCR-amplified FUGW_blasticidin vector backbone using Gibson assembly. Sequences encoding the ALFA epitope tag were engineered in the primers to C-terminally tag the proteins. To generate the long PRPSAP1 isoform, sequences encoding the additional 29 amino acids were ordered as gBlocks gene fragments from IDT and subcloned into FUGW_PRPSAP1-ALFA vector using Gibson assembly. The short isoform of PRPSAP2 was subcloned from FUGW_PRPSAP2-ALFA vector using Gibson assembly. To create the PRPSAP1(1−95)_PRPSAP2-ALFA construct, the first 95 amino acids of PRPSAP1 were PCR amplified from the FUGW_PRPSAP1-ALFA template. Using Gibson assembly, this amplified fragment was then ligated with the PCR-amplified FUGW_PRPSAP2-ALFA construct, which had its first 95 amino acids from PRPSAP2 removed. Other chimeric mutants were generated using similar Gibson assembly approach. *Branchiostoma lanceolatum* Class II PRPS cDNA (Genebank ID# CAH1247366.1) was designed and purchased from Twist Bioscience as synthetic gene fragments and cloned into FUGW_blasticidin vector backbone using HiFi DNA Assembly. Site Directed Mutagenesis was performed to generated PRPSAP1 construct lacking NHR.

Lipofectamine 3000 Transfection Reagent (Invitrogen #L3000001) was used to transfect NIH3T3 cells according to manufacturer's instructions.

### Virus production and transduction

Retrovirus was produced in HEK293T cells by co-transfecting the retroviral transfer vector with pUMVC (Addgene #8449) and pMD2.G (Addgene #12259). For lentivirus production, the lentiviral transfer vector was used in conjunction with psPAX2 (Addgene #12260) and pMD2.G (Addgene #12259). Transfection was carried out using PolyFect transfection reagent (QIAGEN #301105), and the media was changed after 24 h. The supernatant containing the virus was collected at 48 h and 72 h post-transfection, filtered, and concentrated using Retro-X (Takara #631456) or Lenti-X (Takara #631232) concentrators. The resulting viral pellet was resuspended in fresh cell growth media and added to target cells. After 24 h of infection, the virus was removed, and the cells were provided with appropriate selection media.

### Mice

All mice used in the course of the work were cared for in accordance with the University of Cincinnati's institutional guidelines. All

procedures were performed according to the protocols approved by the Institutional Animal Care and the Use Committee under protocol no. 21-04-16-02. Male C57BL/6 mice, 12 weeks of age, were used to extract tissues for size exclusion chromatography (SEC) experiments.

## Tissue protein extraction
After anesthesia, the mice were transcardially perfused with PBS through the ventricular catheter. Organs/tissues were harvested and snap frozen in liquid nitrogen, then homogenized using a mortar and pestle. For western blotting, samples were lysed with RIPA buffer (Thermo #89901) containing 1× protease and phosphatase inhibitor cocktail (Thermo #78446) while sample for SEC experiments were lysed in non-denaturing lysis buffer (50 mM Tris-Cl, pH 7.5, 200 mM NaCl, 1% digitonin, 1 mM TCEP, 1 mM MgCl$_2$, benzonase and 1× protease and phosphatase inhibitor cocktail).

## SDS-PAGE and Western blotting
For denaturing cell lysis, cells were first rinsed once with ice-cold PBS and then lysed in RIPA buffer (Thermo Scientific #89901) supplemented with 1× protease and phosphatase inhibitor cocktail (Thermo #78446). The cleared protein lysates were subsequently mixed with 1× Laemmli sample and separated on either 10% or 12% TGX Fastcast gels (BioRad #1610173 and #1610175). The proteins were then blotted onto 0.2 μm PVDF membrane using Bio-Rad Trans-Blot Turbo Transfer system. PVDF membranes were blocked for 40 min at room temperature (RT) with 5% (w/v) milk in Tris-Buffered Saline (TBS) with 0.1% Tween-20 (TBS-T). After blocking, the membranes were washed and incubated overnight at 4 °C with primary antibodies (diluted 1:1000) prepared in 3% BSA in TBS-T. Primary antibodies used were: CAD (Cell Signaling #93925), TCP1-η (Santa Cruz #sc-271951), FASN (Cell Signaling #3180), FLC (Santa Cruz #sc-390558), HK2 (Cell Signaling #2867), AK2 (Santa Cruz #sc-374095), PRPS1/2 (Santa Cruz #sc-100822), PRPS1 (Proteintech #15549-1-AP), PRPS2 (Sigma #SAB2107995), PRPS1/2/3 (Santa Cruz #sc-376440), PRPSAP1 (Santa Cruz #sc-398422), PRPSAP2 (Proteintech #17814-1-AP), HSP90 (Cell Signaling #4877), β-Actin (Cell Signaling #4970; Cell Signaling #3700), ALFA-HRP (SynapticSystems # N1505-HRP), XO (Abcam #109235), Ras (G12V Mutant Specific) (Cell Signaling #14412), Phospho-p44/42 MAPK (Erk1/2) (Thr202/Tyr204) (Cell Signaling #4376), p44/42 MAPK (Erk1/2) (Cell Signaling #9102), β-Tubulin (Cell Signaling #2128), Phospho-AMPKα (Thr172) (Cell Signaling #2535), AMPKα (Cell Signaling #2532), cleaved PARP1 (Abcam #32064), GAPDH (Cell Signaling #5174), HPRT (Abcam #109021).

Subsequently, the membranes were washed again and incubated with corresponding secondary antibodies from Jackson ImmunoResearch (diluted 1:25000) in 5% (w/v) milk in TBS-T. Secondary Antibodies used were: Anti mouse (Jackson ImmunoResearch #115-035-003), Anti rabbit (Jackson ImmunoResearch #111-035-003). Blots were visualized using chemiluminescent substrates from Thermo on ChemiDoc Imaging System (BioRad #12003153). Blots were analyzed with Image Lab Software 5.2.1 from Bio-Rad. To facilitate re-probing, Restore PLUS Western Blot Stripping Buffer (Thermo #46430) was used to strip the blots.

## GFP immunoprecipitation assay
Cells stably expressing GFP tagged proteins were harvested and cell pellets were lysed in non-denaturing lysis buffer (50 mM Tris-Cl, pH 7.5, 200 mM NaCl, 1% digitonin, 1 mM TCEP, 1 mM MgCl$_2$, benzonase and 1× protease and phosphatase inhibitor cocktail) for 20 min on ice. The cell lysates were then clarified by centrifugation (15,000 × $g$, 15 min, 4 °C) and incubated with the equilibrated anti-GFP nanobody conjugated to magnetic particles (Chromotek GFP-Trap® Magnetic Particles M-270) for 1 h on a rotator disk at 4 °C. The supernatant was removed, and the beads were washed four times in wash buffer (10 mM Tris-Cl, pH 7.5, 1.5 M NaCl, 0.5% Triton X-100, 0.5 mM EDTA). The beads were then boiled for 10 min at 95 °C in 2× SDS-sample buffer for

protein elution. The eluates were sent to University of Cincinnati Proteomics Laboratory for mass spectrometry (MS) analysis.

## ALFA-tag pulldown assay
NIH3T3 cells endogenously expressing PRPS1-ALFA[84] proteins were lysed using non-denaturing lysis buffer (50 mM Tris-Cl, pH 7.5, 200 mM NaCl, 1% digitonin, 1 mM TCEP, 1 mM MgCl$_2$, benzonase and 1× protease and phosphatase inhibitor cocktail) for 20 min on ice. The cell lysates were then clarified by centrifugation (15,000 × $g$, 15 min, 4 °C) and incubated with the equilibrated anti-ALFA nanobody conjugated to agarose beads (ALFA SELECTOR ST, Nanotag #N1511) for 1 h on a rotator disk at 4 °C. The supernatant was removed, and the beads were washed four times in wash buffer (25 mM Tris-Cl, pH 7.5, 0.5 M NaCl, 0.5% Triton X-100). Finally, the beads were boiled for 10 min at 95 °C in 2× SDS-sample buffer for protein elution.

## Size exclusion chromatography
Cells or tissues were lysed using non-denaturing lysis buffer (50 mM Tris-Cl, pH 7.5, 200 mM NaCl, 1% digitonin, 1 mM TCEP, 1 mM MgCl$_2$, benzonase and 1× protease and phosphatase inhibitor cocktail) for 20 min on ice. The cell lysates were then clarified by centrifugation at 15,000 × $g$ for 15 min at 4 °C and subsequently filtered using a 0.22 μm. About 200 μg of cell lysates were loaded onto either a Superose 6 Increase 3.2/300 column (GE Healthcare #29-0915-98) at the flow rate of 0.04 mL/min or Yarra 3μm SEC-2000 at the flow rate of 0.5 mL/min (Phenomenex #00H-4512-K0) using Thermo Vanquish UHPLC. After passing through the void volume, the sample fractions were collected and concentrated using a 3 K MWCO filter (Thermo #88512) and analyzed via western blotting. Gel filtration calibration kits (GE #28-4038-41, GE #28-4038-42, Sigma #MWGF200-1KT) were used to monitor column performance over time. Different internal standards were probed for molecular weight calibration: CAD (GLN-dependent carbamoyl phosphate synthetase (CPS-2), aspartate transcarbamoylase (ATC), and dihydroorotase (DHO))[85], TCP-1η (T-Complex Protein 1 subunit eta)[86], FASN (Fatty Acid Synthase)[87], FLC (Ferritin Light Chain)[88], XO (Xanthine Oxidase)[89], GAPDH (Glyceraldehyde 3-phosphate dehydrogenase)[90], and HPRT (Hypoxanthine phosphoribosyltransferase)[91] form complexes of around 1500 kDa, 900 kDa, 540 kDa, 480 kDa, 290 kDa, 144 kDa, 100 kDa, respectively, while HK2 (Hexokinase 2) and AK2 (Adenylate Kinase 2) are mostly monomeric at 102 kDa and 27 kDa, respectively. The elution profile of PRPS was compared with that of known globular protein complexes, which were used as internal standards to estimate the size of the PRPS enzyme complex. Internal standards were probed in every SEC run.

## Mass spectrometry analyses
To analyze fractions collected from SEC of HEK293T cells, cells were first lysed using non-denaturing lysis buffer (50 mM Tris-Cl, pH 7.5, 200 mM NaCl, 1% digitonin, 1 mM TCEP, 1 mM MgCl$_2$, benzonase and 1× protease and phosphatase inhibitor cocktail) for 20 min on ice. The cell lysates were then clarified by centrifugation at 15,000 × $g$ for 15 min at 4 °C and subsequently filtered using a 0.22 μm. About 200 μg of cell lysates were loaded onto Bio SEC-5 2000Å (Agilent #5190-2543) at the flow rate of 0.2 mL/min using Thermo Vanquish UHPLC. Samples were pooled from the indicated fractions and concentrated with a 10 K MWCO filter (Thermo #88517). Subsequently, the samples were dried in a speed vacuum centrifuge and resuspended in TEAB buffer according to a standard in-solution digestion protocol. The samples were reduced with TCEP (tris-(2-carboxyethyl) phosphine) followed by alkylation with methyl methanethiosulfonate. The samples were digested overnight at 37 °C, which was stopped by adding 10% formic acid. After drying, the samples were resuspended in 20 μl of 0.1% formic acid. A quarter of each sample was then subjected to nanoLC-MS/MS analysis (Orbitrap Eclipse). Peptide fragmentation spectra were

searched against the human database using Proteome Discoverer version 2.4 and the Sequest HT search algorithm (Thermo).

For the analysis of eluates from GFP IP, 30 μL of each eluate sample was loaded onto Invitrogen 4–12% Bis-Tris gels and separated using MOPS buffer. Pre-stained MW marker was used between lanes to facilitate cutting out the full protein region for each sample. The bands were excised, reduced with DTT, alkylated with IAA, and digested overnight with trypsin. The resulting peptides were then extracted, dried, and resuspended in 7 μL of 0.1% formic acid. Following centrifugation at $10,000 \times g$ to remove particulates, 5.5 μL of each sample was analyzed using nanoLC-MS/MS (Orbitrap Eclipse). Peptide fragmentation spectra were searched against the human and mouse database using Proteome Discoverer version 2.4 and the Sequest HT search algorithm (Thermo).

### CRISPR/Cas9-mediated genome editing

PRPS1 (P1) and PRPS2 (P2) KO clones were created in NIH3T3 cells using a CRISPR/Cas9 nickase system. To clone the crRNAs−BfuAI restriction sites, tracrRNA, and GFP sequences were added into a pLKO.1−TRC cloning vector. PRPS1 and PRPS2 crRNA oligonucleotides were inserted to a BfuAI-digested pLKO.1 backbone. These plasmids, containing the sgRNA and Cas9 D10A Nickase (Addgene #41816), were co-transfected into NIH3T3 cells. After 48 h, cells were clonally sorted based on GFP expression onto collagen-coated 96-well plates using BD FACSAria. Clones were assayed for loss of protein expression by Western blotting and were further checked for mutations by purifying genomic DNA and performing PCR on the region spanning the edit site.

For PRPSAP1 (AP1) and PRPSAP2 (AP2) KO clones, NIH3T3 cells were first transduced with a dox-inducible lentiviral Cas9 plasmid (Addgene #50661) and selected with puromycin. Annealed crRNAs were cloned into a lenti-sgRNA vector with hygromycin selection (Addgene #104991) using BsmBI digestion. These lentiviral sgRNA vectors were then introduced into cells expressing dox-inducible Cas9 and selected with hygromycin. Cells were treated with 1 μg/mL doxycycline for a week before clonally sorting them onto collagen-coated 96-well plates using BD FACSAria. Clones were validated for loss of protein expression by Western blotting.

Lentiviral AP2 sgRNA vectors were introduced into validated AP1 KO cell lines to create AP1/AP2 double KO cell lines. P1 and P2 sgRNAs were cloned into a lenti-sgRNA vector with neomycin selection (Addgene #104992) using BsmBI digestion. These sgRNA vectors were then introduced in AP1, AP2 and/or AP1/AP2 KO cell lines to generate the remaining double (AP1/P1, AP1/P2, AP2/P1, AP2/P2) and P2/AP1/AP2 triple KO cell lines. Cells were selected with neomycin and treated with 1 μg/mL doxycycline for a week before clonally sorting them onto collagen-coated 96-well plates using BD FACSAria. Clones were validated for loss of protein expression by Western blotting. All the sgRNAs used are listed in Supplementary Data 8. CRISPR/Cas9 and a DNA donor repair template were used to generate endogenous PRPS1-mNG11-ALFA knock-in via homology-directed repair. Plasmid donor DNA was created by inserting two homology arms (~800 bp each) flanking the exonic mutant sequences into an empty mammalian expression vector (Addgene #68375) using Gibson assembly. PRPS1 crRNA (IDT) was annealed with tracrRNA (IDT) to form sgRNA and incubated with Cas9 protein (IDT) to form a ribonucleoprotein complex (RNP). The RNP mixture, along with the donor plasmid, was electroporated into NIH3T3 cells using the Neon Transfection System (Invitrogen #MPK5000) according to the manufacturer's instructions. Forty-eight hours after electroporation, cells were transfected with pSFFV_mNG3K (1–10) (Addgene #157993) that expresses the first ten beta strands of the monomeric neon green fluorescent protein. Thirty-six hours after transfection, correctly knocked-in cells were clonally sorted based on neon green fluorescence protein expression. ALFA expression was confirmed using immunoblots. For further validation, genotyping was performed by sequencing PCR amplicons containing the edit site. All the oligos and primers used are listed in Supplementary Data 8.

### HPA dataset

The Human Protein Atlas (HPA)[92] tissue consensus database was examined to explore the transcript level of PRPS complex members across various human tissues. The consensus normalized expression levels (nTPM) value for each gene and tissue type represents the maximum nTPM value based on data from both HPA and Genotype-Tissue Expression (GTEx).

### Cell cycle analysis

The cells were harvested and washed in ice-cold PBS followed by fixation in 66% ice-cold ethanol solution. Subsequently, the cells were stained with Propidium Iodide (Sigma-Aldrich #P4864) at a final concentration of 50 μg/mL and RNase (Roche #111119915001) at a final concentration of 50 μg/mL. Samples were then incubated in the dark in a 37 °C incubator prior to analysis on BD LSRFortessa. Data was analyzed using FlowJo (BD Biosciences) software, and profiles and cell cycle distribution generated using the Watson (pragmatic) model. A figure exemplifying the gating strategy is provided as a Source Data file.

### Seahorse assays

Adherent cells were seeded at a density of 10,000 cells/well in DMEM supplemented with 10% FBS and 1× penicillin/streptomycin in a Seahorse XF96 Cell Culture Microplate. The following day, the media was changed to DMEM (Agilent #103575-100) supplemented with 10 mM glucose, 2 mM glutamine, and 1 mM pyruvate and incubated in a 37 °C non-CO₂ incubator for 1 h before the assay. Seahorse XF Cell Mito stress Test (Agilent #103015-100) and XF ATP Rate Assay (Agilent #103592-100) were performed according to the manufacturer's instructions on a Seahorse XFe96 Analyzer (Agilent #S7800B). In both assays, the last injection port was utilized for injecting Hoechst stain. Hoechst fluorescent intensity was measured using a CLARIOstar microplate reader and used to normalize for cell number. Data were exported and analyzed using the Seahorse Wave Desktop Software (Agilent).

### ATP determination assay

Total cellular ATP was measured on a CLARIOstar microplate reader using ATP Determination Kit (Invitrogen #A22066) following manufacturer's instructions. Protein levels were quantified using BCA reagent (Thermo #23227).

### Soft agar colony formation assay

Cells were seeded at a density of 10,000 cells per well in the top layer of 0.3% agar (Lonza SeaKem LE Agarose #50000) mixed with culture media (Gibco DMEM #12800-058), supplemented with 10% FBS and 1× penicillin/streptomycin, in 6-well plates. The bottom layer consisted of 0.6% agar with the same culture media composition. The top layer was replenished weekly, and images were captured after 3.5 weeks. Image quantification was performed using the Trainable Weka Segmentation Plugin in FIJI[93]. Colonies larger than 9000 μm² were classified as true positives, based on the area of colonies observed in non-transformed NIH3T3 cells. Given the average surface area of an NIH3T3 cell, a cumulative area of 9000 μm² represents at least 35 cells.

### Cell collection and processing for metabolomics

For the stable isotope experiment, NIH3T3 fibroblasts were seeded in 15 cm plates and cultured in DMEM. The media were replaced with glucose-free media containing 10 mM of $^{12}C_6$-glucose (unlabeled) or $^{13}C_6$-glucose (Cambridge Isotope Laboratories), supplemented with 10% dialyzed FBS and 1× penicillin/streptomycin, and incubated at

37 °C for 30 min or 5 h. After 30 min or 5 h of isotope exposure, the medium was aspirated, and the cells were washed 3 times with ice-cold PBS. Subsequently, metabolic activity was halted by quenching with ice-cold acetonitrile ($CH_3CN$), followed by the addition of nanopure water ($CH_3CN$:$H_2O$ at 2:1.5 (V/V)) to facilitate cell scraping and collection. Polar and non-polar metabolites were extracted using the solvent partition method−acetonitrile: water: chloroform ($CH_3CN$:$H_2O$:$CHCl_3$ at 2:1.5:1 (V/V))[94]. The aqueous phase containing polar metabolites and the organic phase containing non-polar metabolites were separated and dried using vacuum lyophilization (CentriVap Labconco) or a SpeedVac device. The protein pellets were washed in 500 μL of 100% methanol and centrifuged at $12,000 \times g$ at 4 °C for 10 min. After discarding the supernatant, the protein pellet was dried in a SpeedVac centrifuge for 20 min. The resulting protein residue pellet was used for normalization.

### NMR data acquisition, processing and analysis

For the analysis of intracellular metabolites, the lyophilized polar extracts were resuspended in 220 μL of NMR buffer containing 100 mM phosphate buffer (pH 7.3), 1 mM trimethylsilyl propionic acid-$d_4$ sodium salt (TSP) as internal standard, and 1 mg/mL sodium azide in 100% deuterium oxide ($D_2O$). Two hundred microliter of each sample was transferred to a 3 mm NMR tube. One-dimensional (1D) [1]H-NMR spectra were acquired at 288 K on a Bruker Avance III HD 600 MHz spectrometer (Bruker Biospin) equipped with a 5 mm Broad Band Observed Prodigy probe. The noesygppr1d pulse sequence was employed with water presaturation (25 Hz bandwidth), 512 transients, a 15-ppm spectral width, a 4.0 s relaxation delay, and a 2.0 s acquisition time. Before Fourier transformation, spectra were zero-filled to 128 K data points and apodized with a 1 Hz exponential line-broadening function. Additionally, 1D [1]H-[13]C Heteronuclear Single Quantum Correlation (HSQC) spectra were recorded using the hsqcetgppgsisp2.2 pulse sequence with a 15-ppm spectral width, 1024 transients, 1.75 s relaxation delay, and a 0.25 s acquisition time. The spectra were processed with zero-filling to 16 K data points and apodized with unshifted Gaussian function and 4-Hz exponential line broadening. All spectra were recorded and transformed using Topspin 3.6.2 software (Bruker BioSpin, USA) and processed (phased and baseline corrected) using MestReNova software (MNova v12.0.3, Spain). The spectra were internally calibrated to the methyl resonance of the TSP at 0.0 ppm. Metabolites were identified and assigned by comparing with in-house databases, public databases, Human Metabolome database[95], Biological Magnetic Resonance Data Bank[96], and literature reports. Additionally, 2D [1]H-[1]H Total Correlation Spectroscopy experiments were recorded to facilitate the identification of biochemical substances.

To determine the metabolite abundance and their [13]C isotopomers across the samples, the area of each assigned and well resolved metabolites were manually integrated using global spectra deconvolution, a line-fitting deconvolution algorithm available in MestReNova software (MNova v12.0.3, Spain), which returns the area of each peak of interest. The peak area was divided by the number of protons contributing to that signal. For absolute quantification the corrected peak areas were converted to molar concentration by calibration against the peak intensity of TSP at 0 ppm for [1]H spectra and that of lactate methyl group resonance at 1.32 ppm (quantified from 1D [1]H spectra) for 1D [1]H-[13]C-HSQC spectra before normalization with milligrams (mg) of protein residue in each sample as a proxy of cell amount.

### Statistics and reproducibility

All statistical analyses were performed using GraphPad Prism software (GraphPad Software). Sample sizes, replicates, and statistical tests used are indicated in each figure legend. For all statistical analyses, $P$ values are indicated in each corresponding figure.

## Data availability

All data supporting the findings of this study are available within the Article and its Supplementary Information, which includes the mass spectrometry proteomics data, the NMR metabolomics data, and the sgRNAs used for generating CRISPR knock-in/knock-out cell lines. For evolutionary analyses, the datasets (sequences; genomic, transcriptomic, or amino acid) generated and/or analyzed in this study have been deposited to Figshare and available at https://doi.org/10.6084/m9.figshare.27146256. The NCBI accession codes used in our analyses are listed in Supplementary Data 1 and 2 as well as in the corresponding FASTA files in the Figshare datasets. All accession codes refer to entries in the National Center for Biotechnology Information (NCBI) and can be accessed at http://www.ncbi.nlm.nih.gov. The mass spectrometry proteomics data generated in this study have been deposited to the ProteomeXchange Consortium via the PRIDE partner repository under accession codes PXD058828, PXD058829, and PXD059156. The metabolomics data generated in this study have been deposited to the NIH Common Fund's National Metabolomics Data Repository−the Metabolomics Workbench under accession code PR002247 (https://doi.org/10.21228/M85V6B). Source data are provided with this paper. All unique reagents generated in this study will be made available by the corresponding author upon request with a completed Materials Transfer Agreement. Source data are provided with this paper.

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

## Acknowledgements

This work received support by NIH grants (R01CA230904 and R35GM133561) to J.T.C. and S10OD026717 to K.D.G. for the mass spectrometer used in the proteomics studies. We thank the Translational Metabolomics Facility at Cincinnati Children's Hospital Medical Center, and the University of Cincinnati Proteomics laboratory for their assistance. We thank D. Plas, K. Patra, A. Waters, C. Bartolacci, and J. Meller

for their insights during the manuscript review process. The silhouette images in phylogenetic trees were downloaded from Phylopic (http://phylopic.org/) or designed using Adobe Illustrator (Adobe, USA). All images downloaded were freely available for reuse under a Public Domain license. Schema generation and figure formatting was done using Adobe Illustrator (Adobe, USA).

## Author contributions
Conceptualization, B.R.K. and J.T.C.; Methodology, B.R.K. and J.T.C.; Investigation, B.R.K., J.T.C., A.C.M., S.V.M.; Formal analysis, B.R.K., A.C.M., S.V.M.; Writing–Original Draft, B.R.K. and J.T.C.; Writing–Review & Editing, B.R.K., J.T.C., A.C.M., S.V.M., K.D.G., L.E.R.; Visualization, B.R.K. and J.T.C.; Project administration, J.T.C.; Supervision, J.T.C., L.E.R., K.D.G.; Funding Acquisition, J.T.C.

## Competing interests
B.R.K. and J.T.C. have filed patent applications on this work. The remaining authors declare no competing interests.
