## [Transparent Peer Review file · Nature Communications]

The role of gene duplication and paralogs specialisation in the evolution of the mammalian PRPS complex

Corresponding Author: Dr John Cunningham

Version 0:

Reviewer comments:

Reviewer #1

(Remarks to the Author)

In the submitted manuscript Karki and colleagues describe the evolution of the mammalian PRPS enzyme complex by tracing the existence of its multimers in various organisms from opisthokonts to mammals. Multiple homologues of PRPS polypeptides exist in mammals – PRPS1, PRPS2 and the testis-specific PRPS1L1 – in addition to PRPS-associated gene products, PRPSAP1 (AP1) and PRPSAP2 (AP2). Three classes of PRPS enzymes are defined on their regulation.

Comparison of the evolutionary pattern of Class I and II homologues was consistent with their co-evolution, although Class II homologues were lost in Holomycota. This loss seems to be compensated by duplication of Class I PRPS homologues allowing them to acquire non-homologous, in frame insertions, designated NHRs. Specifically, these NHRs are in the regulatory and catalytic regions in two of the five PRPS proteins present in *Saccharomyces cerevisiae*, rendering them catalytically inactive and explaining the requirement of heterodimers for PRPP synthesis in this yeast.

By means of extensive homology comparisons of PRPS gene products the evolution of a biochemical complex capable of both functional and regulatory features has been identified from PRPS1 to PRPS2 and AP1 from AP2, respectively.

The manuscript is well-written and the comprehensive homology comparisons provide a convincing argument for the creation of the protein complexes responsible for the provision of the essential metabolite PRPP in mammals. These studies carried out in various tissues are supported by IP assays and SEC and confirm the multimeric nature of mammalian PRPS enzymes, extending the fundamental studies by M Tatibana and colleagues.

Intra-complex cooperation is shown to be essential for optimal PRPS activity in mammalian cells by creating viable knock-out combinations which demonstrated the flexible nature of the PRPS complex for the cell type adaptation of metabolism. This combinatorial approach is convincingly illustrated in Figure 5 (H) for various PRPS complexes.

Overall, the manuscript provides an impressive set of comprehensive data for the PRPP-synthetic activity in mammalian cells, comprising complex formation, topology and their impact on activity and metabolic output – a veritable 'tour-de-force'.

Major comment:

I found the abstract did not mirror the important findings reported in the manuscript, although I am aware of the word limit.

Minor comments:

1. Legend of Figure 1 (B). The red and black dots were visible only on screen.
2. Figure 1 (C) Colour coding in heading?

3. Re-checking some of the positions of NHRs in Figure 2 (E).

Reviewer #2

(Remarks to the Author)

This manuscript describes how a single phosphoribosyl pyrophosphate synthetase (PRPS) gene evolved into a multicomponent enzyme complex involving multiple homologs. The authors trace the evolutionary history across eukaryotes, showing that duplications and losses of PRPS homologs led to a large, heterogeneous complex that dynamically adjusts activity. Using genetic knockouts in cell lines, they demonstrate that no single PRPS enzyme alone can sustain normal growth and metabolism, revealing how specific dimer-pairing rules and translational control of different PRPS subunits create emergent regulatory properties. These findings provide a framework for understanding how evolutionary innovations in PRPS genes and complex assembly result in control over nucleotide biosynthesis and by extension broader metabolic flux.

While the manuscript overall is well written and presented, I have several concerns that may warrant substantive revisions. Following, I list these concerns, mostly in the order they appear in the text:

L45 (also L178-180): The authors should expand on the logic of their conclusion that, "multiple bacterial species must have contributed these key paralogous metabolic enzymes to LECA's biochemical framework."

***67: The catalog of PRPS enzymes cannot "illustrate[s] how selective pressures contribute to the emergence of new properties over evolutionary timescale".

L70: "newly annotated sequences" - more information about the extent of whole-genome sampling would be useful here. An important concern in all studies of gene family evolution is, to what extent is there evidence for absence, versus simply having incomplete sequence/assembly/annotation across the taxa surveyed. The search methods described in L438-446 do not provide enough information to know how much evidence for absence they might have. Just blasting against NCBI provides little to no evidence for absence given the limited utility of SRA for this kind of search, plus the ubiquity of poor assemblies and annotations. A thorough enough investigation of at least some subset of whole genomes to identify pseudogene presence/absence, might provide more confidence that homologs are not being missed.

****L72 We hypothesized that this increased repertoire of PRPS homologs imbues eukaryotes with enhanced metabolic adaptability by virtue of additional regulatability and increased biosynthetic capacity.****

L75: "Given the well-established evolutionary trajectory from opisthokonts to mammals..." It is not clear what trajectory the authors are referring to. Are they talking about the analysis behind Figure 1 or based on the literature (there are no citations)? Is it just the trajectory of increased number of PRSP genes or something broader?

L88-92 & Fig 1B: Is the red arrow at the top of Fig 1B pointing to the "highly conserved" splice site junction? If so, I'm confused because that junction isn't even between the same two exons in many cases.

Figure 1: Overall, Figure 1 is well done and very helpful, but in addition to my question about the splice site junction conservation, I have a couple other questions/suggestions. 1) The open box legend says "mostly present". I initially interpreted this as the whole gene is not conserved across Teleosts, but upon reflection decided the authors probably mean "Present in most species". Perhaps restate this in the legend to avoid that potential confusion. 2) In panels B, D, and E, I assume exons are color coded by sequence/functional homology, but it is not stated in the caption and the fact that red and black dots, in the regulatory and catalytic domains respectively, occur in different exons across taxa makes me question this assumption. Perhaps some type of indicator for which exons make up the regulatory and catalytic domains would make interpretation more clear.

L92-94 & L105-117: The authors should discuss how well (or whether) these ideas fit with Ohno's 2R hypothesis about whole genome duplication (WGD) events near the base of vertebrates, which is well-established by whole-genome analyses (e.g. Yu, et al. 2024. *Nat Ecol Evol* 8, 519–535). It seems like the vertebrate expansion of PRSPs may be another example of what is seen in many gene families (most famously the Hox genes). This discussion would make the study of interest to additional researchers investigating the broad consequences of WGD. On L377 the authors refer to them as "independent gene duplications", but are they really independent?

One way to potentially address their hypothesis about Class II loss being mitigated by Class I gain (L105-117, L402ff) would be to see if they could find pseudogene remnants of the Class II PRPS genes in Holomycota and analyze whether the timing of pseudogenization coincides with Class I gains. The presence of Class II in Chytridomycota suggests the gain of Class I paralogs may considerably predate widespread loss of Class II, so perhaps the gain/loss are not as well associated as the "compensatory selection" (L116) argument suggests.

L103-104: There is a fairly extensive literature about reduplicated genes having a bias toward expression in testes that makes even the "speculation" that PRPS1L1 plays a role in testes phenotypic evolution not well-supported by the analysis/evidence presented in this paper. It's still interesting, but only as another example that fits a much wider trend (e.g. N. Vinckenbosch, et al. 2006. *PNAS* 103 (9) 3220-3225; Wang, et al. 2012. *J Mol Evol* 74, 113–126).

L130: The statements about strength of selection and/or pace of adaptation are not supported by functional analyses. At best, the functional data "suggest" the changes were adaptive.

L181-188: It's not clear to me what this molecular weight analysis is supposed to inform relative to the hypotheses of the study.

L366: this is the first/only mention of symbiosis and genetic reshuffling. The authors should explain how they arrive at this inference.

L447 The method for intron/exon boundary detection lacks sufficient detail. (what does "sourced" mean in this context). Also, need a coma after "available" on line 449.

Minor issues:

Throughout the text the authors use "highly homologous" to mean "highly simliar" or maybe "high sequence identity". Genes/sites/positions are either homologous or not; homology is a state not a continuous trait.

L420 delete "its"

Version 1:

Reviewer comments:

Reviewer #2

(Remarks to the Author)

The authors have done an excellent job addressing the concerns raised in my initial review. They have clarified key points regarding the inheritance of Class II PRPS enzymes, improved transparency through expanded and revised tables and figures, and appropriately tempered previous overstatements about selective pressures and adaptation. I particularly appreciate the nuanced revisions that acknowledge the limitations of their data and incorporate relevant literature, including the connection to Ohno's 2R hypothesis (though I note Ohno's original work remains uncited, why?) and broader trends in retroduplication.

The manuscript is now significantly clearer and more rigorous, particularly in its treatment of evolutionary inference and gene family expansion. The authors' responses are thoughtful and thorough, and the revised manuscript represents a valuable contribution to our understanding of PRPS gene evolution in eukaryotes. I am satisfied that my concerns have been fully addressed.

RESPONSE TO REVIEWERS' COMMENTS

Please note that the line numbers referenced in the responses below correspond to the “No Markup” view in the Track Changes options. For convenience, all updated text in the manuscript has been highlighted.

Reviewer #1 (Remarks to the Author):

In the submitted manuscript Karki and colleagues describe the evolution of the mammalian PRPS enzyme complex by tracing the existence of its multimers in various organisms from opisthokonts to mammals. Multiple homologues of PRPS polypeptides exist in mammals – PRPS1, PRPS2 and the testis-specific PRPS1L1 – in addition to PRPS-associated gene products, PRPSAP1 (AP1) and PRPSAP2 (AP2). Three classes of PRPS enzymes are defined on their regulation.

Comparison of the evolutionary pattern of Class I and II homologues was consistent with their co-evolution, although Class II homologues were lost in Holomycota. This loss seems to be compensated by duplication of Class I PRPS homologues allowing them to acquire non-homologous, in frame insertions, designated NHRs. Specifically, these NHRs are in the regulatory and catalytic regions in two of the five PRPS proteins present in *Saccharomyces cerevisiae*, rendering them catalytically inactive and explaining the requirement of heterodimers for PRPP synthesis in this yeast.

By means of extensive homology comparisons of PRPS gene products the evolution of a biochemical complex capable of both functional and regulatory features has been identified from PRPS1 to PRPS2 and AP1 from AP2, respectively.

The manuscript is well-written and the comprehensive homology comparisons provide a convincing argument for the creation of the protein complexes responsible for the provision of the essential metabolite PRPP in mammals. These studies carried out in various tissues are supported by IP assays and SEC and confirm the multimeric nature of mammalian PRPS enzymes, extending the fundamental studies by M Tatibana and colleagues.

Intra-complex cooperation is shown to be essential for optimal PRPS activity in mammalian cells by creating viable knock-out combinations which demonstrated the flexible nature of the PRPS complex for the cell type adaptation of metabolism. This combinatorial approach is convincingly illustrated in Figure 5 (H) for various PRPS complexes.

Overall, the manuscript provides an impressive set of comprehensive data for the PRPP-synthetic activity in mammalian cells, comprising complex formation, topology and their impact on activity and metabolic output – a veritable ‘tour-de-force’.

We thank Reviewer 1 for the kind words and careful reading of our manuscript as well as the suggested improvements that will undoubtedly enhance clarity.

Major comment:

I found the abstract did not mirror the important findings reported in the manuscript, although I am aware of the word limit.

We tried to capture the full breadth of our study within the word limit constraints of the abstract. To achieve this, we had to over-summarize important details regarding PRPS complex organization, assembly, activity, and regulation (and when, why, and how they originated), which are all addressed to varying degrees within the main text of the paper. Being self-critical, we can identify several key details given short shrift in the abstract including the observation of tissue-specific differences in PRPS complex architecture, the inadequacy of PRPS1 as a standalone homomeric enzyme, the association between losses in Class II PRPS enzymes with gains in the class I PRPS enzyme repertoire, etc. From a practical standpoint, it was just untenable to adequately summarize all the “major” findings, which might be different depending on the reader’s purview.

Minor comments:

1. Legend of Figure 1 (B). The red and black dots were visible only on screen.

We appreciate the reviewer's feedback. We have now removed the red and black dots that were previously shown only in the PRPS1-encoding gene structures. To improve consistency and clarity, we have replaced them with red and black dotted lines throughout all PRPS homologs presented in Figure 1B, indicating the catalytic and flexible loop regions in the corresponding homologs (see updated Figure 1B). Additionally, we have included a new Supplementary Table 3, which details the exact amino acids spanning the catalytic and regulatory flexible loops. The amino acids translated from the corresponding exons are color-coded according to the schema in Figure 1B. Furthermore, the specific amino acids comprising the insertions in PRPSAP2 and Prs5, represented by hatch marks in Figure 1B, have been underlined in the table for clarity.

2. Figure 1 (C) Colour coding in heading?

We thank the reviewer for your observation. In Figure 1A, the PRPS homolog status (indicated by a colored rectangular box) is not positioned directly above the rectangles, so we provided a color-coded legend on the left-hand side. However, in Figure 1C, the specific PRPS homolog (color-coded) being described is placed directly above the corresponding rectangular boxes, so additional color coding in the left-hand captions was not included.

3. Re-checking some of the positions of NHRs in Figure 2 (E).

We have carefully re-checked and confirmed the positions of NHRs in Figure 2E. While doing so, the reviewer's comment made us consider ways to further improve the figure's clarity for readers. As a result, we have now explicitly specified which PRPS enzymes were used as references to indicate the insertion points for Prs1, Prs5, PRPSAP1, PRPSAP2, PrsB, and PRPSAP-like homologs. Specifically, we used the ancestral PRPS enzymes—Prs2 for *S. cerevisiae* and PRPS1 for humans—as references. Additionally, we now display the corresponding amino acid positions that span the regulatory and catalytic flexible loops and indicate the number of amino acid insertions (NHRs) present in each NHR-containing PRPS homolog. These updates are reflected in the revised Figure 2E.

Reviewer #2 (Remarks to the Author):

This manuscript describes how a single phosphoribosyl pyrophosphate synthetase (PRPS) gene evolved into a multicomponent enzyme complex involving multiple homologs. The authors trace the evolutionary history across eukaryotes, showing that duplications and losses of PRPS homologs led to a large, heterogeneous complex that dynamically adjusts activity. Using genetic knockouts in cell lines, they demonstrate that no single PRPS enzyme alone can sustain normal growth and metabolism, revealing how specific dimer-pairing rules and translational control of different PRPS subunits create emergent regulatory properties. These findings provide a framework for understanding how evolutionary innovations in PRPS genes and complex assembly result in control over nucleotide biosynthesis and by extension broader metabolic flux.

While the manuscript overall is well written and presented, I have several concerns that may warrant substantive revisions. Following, I list these concerns, mostly in the order they appear in the text:

With profound gratitude, we wish to thank Reviewer 2 for greatly augmenting our manuscript with their extremely helpful suggestions and spot-on criticisms. We have cleaned up the sloppiness in language and improved our manuscript according to the following specific critiques and recommendations.

L45 (also L178-180): The authors should expand on the logic of their conclusion that, "multiple bacterial species must have contributed these key paralogous metabolic enzymes to LECA's biochemical framework."

Eukaryogenesis involved an amalgam of Archaeal genes and Bacterial genes that ultimately provided the metabolic repertoire of the last eukaryotic common ancestor (LECA). Based on our interrogation of Archaeal PRPS enzymes, with a major focus on those within the Asgard lineage thought to be the closest prokaryotic relatives to eukaryotes, we do not find evidence that any currently annotated Archaeal PRPS-encoding genes

are present in any of the eukaryotic genomes/transcriptomes we sampled. Likewise, and now addressed in more fulsome detail (see Supplementary Figure S1A, Supplementary Table 1; figshare), our data show that extant species related to the Alphaproteobacterial endosymbiont that likely contributed the largesse of the ancestral mitochondrial proteome possess only class I PRPS-encoding genes. Although we did not sample all eukaryote genomes/transcriptomes, we did tblastn scan multiple genomes/transcriptomes from multiple taxa within each eukaryotic superphyla (at least where data availability permitted). This non-exhaustive sampling revealed the presence (and prevalence) of Class I PRPS-encoding genes within every major Eukaryotic superphyla, indicating that LECA likely harbored a Class I PRPS-encoding gene. And our analysis revealed the presence and prevalence (albeit to a lesser extent than Class I PRPS) of Class II PRPS-encoding genes within each of the major Eukaryote divisions - Diaphoretickes, Amorphea, Excavata (or Opimoda+ and Diphoda+ - see references #27, #28, #30 and Supplementary Figure S1, Supplementary Table 2). Where possible (Opisthokonts and two branches of Amoebozoa), we now show that Class II PRPS enzymes from some superphyla are more highly related to one another (by amino acid sequence identity) than Class II PRPS enzymes from outgroups, and we show via conserved splice junctions that these orthologs shared a common ancestral Class II PRPS (Figure S4A). This same phenomenon may also hold for Diaphoretickes (or Diphoda+, depending on the reviewers' preferred molecular phylogenetic designation), but we have been far less exhaustive in genomes sampled in those taxa. Fewer currently available Excavata species' genomes/transcriptomes harbor Class II PRPS-encoding genes as compared to the other divisions, and molecular phylogenetic relationships between taxa in this division are less certain in general, even in the newest robustly rooted tree of eukaryotes (reference #30), so it is probably inadvisable to draw qualitative or quantitative conclusions from comparative amino acid sequence analysis between species within this division or between Excavata and Amorphea or Diaphoretickes. Despite this potential caveat, the observation that intron-possessing Class II PRPS enzyme-encoding genes are found in each of the Eukaryotic Divisions, and the amino acid sequence identity is, by-and-large, higher between homologs within division versus between homologs from different divisions best supports a model whereby the major mode of inheritance of Class II PRPS enzymes was through vertical inheritance from a common ancestral Class II PRPS, rather than multiple distinct acquisitions in basal common ancestors of each supergroup.

Additionally, we have not yet identified a bacterial species that harbors both a class I and a class II PRPS-encoding gene. While this absence of evidence does not preclude that one such species (or even an Archaeal species) exists or once existed that could've donated their PRPS-encoding genes to early Eukaryotes, it should be noted that the vast majority of bacterial genomes possess a single PRPS-encoding gene (see Supplementary Tables 1 and 2; figshare). Nevertheless, the caveat remains that we have not been exhaustive, and many species have yet to be sampled or are extinct. Because we cannot be 100% definitive given the limitations expressed above, we have changed the sentence (in L45) to read "multiple bacterial species *likely* contributed these key paralogous metabolic enzymes to LECA's biochemical framework" to reflect the most parsimonious explanation with the requisite level of uncertainty.

***67: The catalog of PRPS enzymes cannot "illustrate[s] how selective pressures contribute 68 to the emergence of new properties over evolutionary timescale".

Reviewer 2 is correct. Our previous statement is confusing and wrong because it is evident that the original change, in the cases we have pinpointed, is the duplication of a gene, genetic region, chromosome, or genome. The possible selective pressure to alter the emergent paralog is inferred based on amino acid sequence changes that occurred rapidly, which indeed influence biochemical properties of that duplicated paralog. So, although the catalog of PRPS enzymes itself does not provide illustrative examples of how selective pressures contributed to emergent properties, it does provide a playground on which selective pressures can (and do) act. We have corrected the sentence to read:

L68-69 ".....curated an extensive catalog of PRPS enzyme sequences that served as a framework for interrogating how selective pressures....."

L70: "newly annotated sequences" - more information about the extent of whole-genome sampling would be useful here. An important concern in all studies of gene family evolution is, to what extent is there evidence for absence, versus simply having incomplete sequence/assembly/annotation across the taxa surveyed. The search methods described in L438-446 do not provide enough information to know how much evidence for absence they might have. Just blasting against NCBI provides little to no evidence for absence given the limited utility of

SRA for this kind of search, plus the ubiquity of poor assemblies and annotations. A thorough enough investigation of at least some subset of whole genomes to identify pseudogene presence/absence, might provide more confidence that homologs are not being missed.

Reviewer 2 makes several very important points, which provide us with excellent opportunities to provide some relevant details that were omitted from the initial submission and supply much-needed clarity to the manuscript. First, we wish to clarify that Figure 1A and accompanying text do not, in and of itself, support vertical inheritance from LECA of Class II, or even the much more prevalent Class I, PRPS enzymes. To support such a claim, the provenance of each Class I or II paralog must be rigorously traced, as we have done with PRPS-associated proteins in Opisthokonts from an ancestral PRPS1 gene (Figure 1B) and with PRPS2 and PRPSAP1 from ancestral vertebrate PRPS1 gene and ancestral vertebrate PRPSAP2 gene, respectively. We have now applied this criterion to class II enzymes from Opisthokonts and Amoebozoa (Discosea and Evosea, where multiple genomes are publicly available), and show a vertical inheritance pattern for these genes in their respective superphyla (Figure S4B and S4C). The reviewer is also correct that the summary of data/sequences, as presented in Figures 1A and 1C, masks or excludes evidence of gene/enzyme absence. We now provide a more exhaustive list of genomes and taxa sampled, including whole genome assemblies where no Class II PRPS-encoding genes/pseudogenes were detected – Supplementary Table 2. The fact that we do not detect pseudogenes in the subset of whole genomes sampled might reflect a model whereby more extensive genomic rearrangements are responsible for the gains and losses of PRPS homologs observed (as in the 2R whole genome duplication event that likely gave rise to PRPS2 and PRPSAP1 genes – see Supplementary Figures S2 and S3). But, more comprehensiveness and granularity will be needed in future studies to pinpoint exact genetic events giving rise to gains and losses in PRPS enzyme repertoires, which includes increased sampling of existing publicly available genomes and evaluation of PRPS ortholog presence/absence in the ever-increasing number of full genomes that will be made publicly available in the future.

****L72 We hypothesized that this increased repertoire of PRPS homologs imbues eukaryotes with enhanced metabolic adaptability by virtue of additional regulatability and increased biosynthetic capacity.****

This statement's purpose (and the hypothesis) reflects our observations that, in addition to the nearly ubiquitous class I PRPS-encoding genes, class II PRPS-encoding genes are also present in all branches of the eukaryotic tree and the Hove-Jensen group (Krath and Hove-Jensen, JBC, 2001) previously demonstrated how class II enzymes possess different regulatory properties and can utilize nucleotides other than ATP as pyrophosphoryl group donors. Additional support for this statement came in the form of identification of the PRPS-associated proteins and the PRPS2 isozyme from the Tatibana and Becker groups, respectively, who performed the initial characterization of these paralogs in vitro.

L75: "Given the well-established evolutionary trajectory from opisthokonts to mammals..." It is not clear what trajectory the authors are referring to. Are they talking about the analysis behind Figure 1 or based on the literature (there are no citations)? Is it just the trajectory of increased number of PRSP genes or something broader?

We thank the reviewer for providing us with an opportunity to expand on the important body of literature that exists on this topic that we couldn't acknowledge due to journal citation limitations. Hopefully the authors of the following key works can forgive their omission. We acknowledge that "trajectory" may not be a perfect word choice, since the following studies are still (and must be) comparative in nature, but nevertheless, they demonstrate a core program shared between basal Opisthokonts and the later-evolving mammals that clearly followed a path that includes expansion, contraction, and diversification of genetic, epigenetic, developmental, and metabolic repertoires. Our work wouldn't have been possible without these previous studies and the roadmap they provided for us. Some seminal studies that we wish to highlight here include robust characterization of Ichthyosporeans: Shabardina, V. et al. Ichthyosporea: a window into the origin of animals. *Commun Biol* 7, 1–13 (2024), and Olivetta, M., Bhickta, C., Chiaruttini, N., Burns, J. & Dudin, O. A multicellular developmental program in a close animal relative. *Nature* 635, 382–389 (2024); identification and attempted phylogenetic placement of Tunicaraptor - Tikhonenkov, D. V. et al. New Lineage of Microbial Predators Adds Complexity to Reconstructing the Evolutionary Origin of Animals. *Current Biology* 30, 4500-4509.e5 (2020), characterization of Syssomonas and Pigoraptor species (that also includes very robust discussion and

background on the “trajectory” to multicellularity) - Tikhonenkov, D. V. et al. Insights into the origin of metazoan multicellularity from predatory unicellular relatives of animals. *BMC Biology* 18, 39 (2020), and the review that we chose to cite for this statement (L78, reference #31) that discusses key theories in the field for how multicellular animals originated/branched from common ancestral unicellular basal Opisthokonts - Ruiz-Trillo, I., Kin, K. & Casacuberta, E. The Origin of Metazoan Multicellularity: A Potential Microbial Black Swan Event. *Annual Review of Microbiology* 77, 499–516 (2023).

L88-92 & Fig 1B: Is the red arrow at the top of Fig 1B pointing to the "highly conserved" splice site junction? If so, I'm confused because that junction isn't even between the same two exons in many cases.

To increase clarity, we have now included Supplementary Table 3 as a supplementary Excel file, which details the exact amino acids spanning the catalytic and regulatory flexible loops. The amino acids translated from the corresponding exons are color-coded according to the schema in Figure 1B to more clearly illustrate the conserved position of the splice junction within the specified PRPS genes across indicated species. Furthermore, the specific amino acids comprising the insertions in PRPSAP2 and Prs5, represented by hatch marks in Figure 1B. As these alignments show, the red arrow highlighting the conserved splice junction indeed occurs between the same position (same amino acids) in the conserved amino acid sequence, but the number of preceding exons is sometimes different between very distantly related species due to genetic events that the current molecular phylogenetic record does not have the power to resolve. This is not entirely unexpected given the 600+ million years that separates Metazoans and unicellular Opisthokonts from their shared common ancestor. To make this conserved placement of the splice junction more obvious and clear for the reader, we have included positions of regulatory flexible and catalytic flexible loop regions as landmarks to orient the reader.

Figure 1: Overall, Figure 1 is well done and very helpful, but in addition to my question about the splice site junction conservation, I have a couple other questions/suggestions. 1) The open box legend says "mostly present". I initially interpreted this as the whole gene is not conserved across Teleosts, but upon reflection decided the authors probably mean "Present in most species". Perhaps restate this in the legend to avoid that potential confusion. 2) In panels B, D, and E, I assume exons are color coded by sequence/functional homology, but it is not stated in the caption and the fact that red and black dots, in the regulatory and catalytic domains respectively, occur in different exons across taxa makes me question this assumption. Perhaps some type of indicator for which exons make up the regulatory and catalytic domains would make interpretation more clear.

We thank the reviewer for this helpful suggestion to improve clarity. For “mostly present”, we are indeed referring to “present in most taxa” within the indicated lineage. We have included the text (L1063) “*Mostly present indicates present in most taxa*” in the figure legend. We have also now indicated sequence/functional homology regions for the regulatory flexible and catalytic flexible loops to help provide orientation landmarks for readers – see updated Figure 1B. Hopefully, these changes substantially improve clarity for the reader.

L92-94& L105-117: The authors should discuss how well (or whether) these ideas fit with Ohno's 2R hypothesis about whole genome duplication (WGD) events near the base of vertebrates, which is well-established by whole-genome analyses (e.g. Yu, et al. 2024. *Nat Ecol Evol* 8, 519–535). It seems like the vertebrate expansion of PRSPs may be another example of what is seen in many gene families (most famously the Hox genes). This discussion would make the study of interest to additional researchers investigating the broad consequences of WGD. On L377 the authors refer to them as "independent gene duplications", but are they really independent? One way to potentially address their hypothesis about Class II loss being mitigated by Class I gain (L105-117, L402ff) would be to see if they could find pseudogene remnants of the Class II PRPS genes in Holomycota and analyze whether the timing of pseudogenization coincides with Class I gains. The presence of Class II in Chytridomycota suggests the gain of Class I paralogs may considerably predate widespread loss of Class II, so perhaps the gain/loss are not as well associated as the "compensatory selection" (L116) argument suggests.

Indeed, our new data is consistent with Ohno's 2R hypothesis, whereby we nominate second whole genome duplication (2R) occurring in the early jawed vertebrate evolution leading to PRPS2 and PRPSAP1 isoforms from PRPS1 and PRPSAP2, respectively (Supplementary Figures S2 and S3). We also tried to find evidence indicative of which genomic event led to the loss of class II PRPS enzymes from vertebrates or their last common ancestor that possessed a class II PRPS-encoding gene. The basic gene synteny analysis we performed left us unable to conclude whether the whole genome duplication (1R) in ancestral vertebrate evolution contributed to

the loss of Class II PRPS. The possibility remains that there is not a strict concomitant occurrence of class II PRPS-encoding gene loss and Class I PRPS-encoding gene gain. The dearth of whole genome sequencing data in Amoebozoa preclude this type of analysis in other lineages.

In our further investigation of Class II PRPS in the Holomycota lineage, we identified only three species harboring Class II PRPS-encoding genes. *Pompholyxophrys punicea* and *Batrachochytrium salamandrivorans* are intronless, suggesting they may have acquired Class II PRPS through horizontal transfer (Supplementary Figure S4A). The third species, *Lithocolla globossa* (*Nuclearia*), may have obtained Class II PRPS through a similar process, but due to incomplete/non-comprehensive genomic data, its intron status remains unconfirmed. Additionally, we found no evidence of Class II PRPS pseudogene remnants in other Holomycota species (Supplementary Table 2). Thus, it remains possible that Class II PRPS was entirely lost in the ancestral Holomycota and subsequently regained via lateral gene transfer in a select few species.

Our reference to the “independent gene duplications” found in the discussion section (now L390) is collectively referring to the numerous independent genetic events that produced PRPSAP-like proteins. These could be part of major genomic changes such as whole genome duplications, or smaller genetic events like retrotranspositions. For some of these, there currently exists no underlying genetic data that would enable us to be more precise (e.g., *Diphylleia rotans*).

L103-104: There is a fairly extensive literature about retroduplicated genes having a bias toward expression in testes that makes even the "speculation" that PRPS1L1 plays a role in testes phenotypic evolution not well-supported by the analysis/evidence presented in this paper. It's still interesting, but only as another example that fits a much wider trend (e.g. N. Vinckenbosch, et al. 2006. PNAS 103 (9) 3220-3225; Wang, et al. 2012. J Mol Evol 74, 113–126).

There is no functional data regarding the utility of the PRPS1L1 gene/enzyme, but the gene does appear to be under some selective pressure to retain its ability to encode an active enzyme, and our proteomics data in HEK293T cells demonstrates it can effectively incorporate into the multimeric PRPS enzyme complex. It is not required for development or fecundity in mice, and its tissue-restricted expression pattern is noted in several studies in different mammalian species. The reviewer is absolutely correct that we present no evidence for the functional utility of PRPS1L1 in testes phenotypic evolution, so we have made this lack of evidence clear in the text.

L111-112: “Although the high sequence identity of PRPS1L1 among Eutherians suggests selective pressure, the physiological role of testes-restricted PRPS1L1 remains unknown.”

L130: The statements about strength of selection and/or pace of adaptation are not supported by functional analyses. At best, the functional data "suggest" the changes were adaptive.

We agree with the reviewer that we do not provide functional analyses to substantiate the claim that “adaptive changes occurred rapidly in the loop regions of PRPSAP2 post-duplication suggesting strong selective pressure against the catalytic function.” However, the changes that did occur, and which prevailed in all extant species include changes in amino acids at the FLAG region, regulatory flexible loop, catalytic flexible loop, and ribose-5-phosphate binding loop that either coordinate substrate (ATP and/or Ribose-5-Phosphate) binding or facilitate catalysis. If there was not a strong selective pressure against retention of the duplicated copy's (PRPSAP2's) catalytic function, one would expect to observe instances where the duplicated ortholog possesses all amino acids necessary for independent catalytic activity, which we could not find. Notably, the examples of convergent evolution we supplied in CRuMs, Amoebozoa, and Apusozoa phenocopy the structure/function effect on the duplicated copy (PRPSAP2) in Opisthokonts (see Supplementary Figures S3, S4, Supplementary Table 4, and the 20 Amoebozoan PrsB sequences provided in figshare).

L181-188: It's not clear to me what this molecular weight analysis is supposed to inform relative to the hypotheses of the study.

This analysis captures the high molecular weight PRPS complex using unbiased proteomics, which we believe adds an additional level of rigor because it helps rule out the possibility of induced (i.e., more transient)

association of PRPS complex components during the affinity capture proteomics workflow. It also adds value in method validation because proteomics experts may wish to probe and compare the methodology or PRPS complex against other known large assemblies of interest, or inquire as to whether compartmentalized (e.g., mitochondrial, ER-localized, etc.) complexes of interest appear using our methodology. This experiment also utilizes Bio SEC-5 2000 Å column (fractionation range– 150 kDa to >10 MDa) which provides a better fractionation range for analyzing high molecular weight protein/protein complexes compared to the Superose 6 Increase 3.2/300 column (fractionation range – 5 kDa to 5 MDa).

L366: this is the first/only mention of symbiosis and genetic reshuffling. The authors should explain how they arrive at this inference.

The mention of symbiosis is a nod to the endosymbiotic cyanobacteria's donation of their class I PRPS to Archaeplastida organisms and of the bacterial species that must have donated their class I (perhaps the alphaproteobacterial endosymbiont) and class II PRPS (non-alphaproteobacterial symbiont) to the first eukaryotic common ancestor, last eukaryotic ancestor, or some intermediary stage in between. As an example, cyanobacterial PRPS is ~70% identical (by amino acid sequence) to various Archaeplastida PRPS sequences (mainly from Viridiplantae, but also Glaucophyta).

Genetic reshuffling is a bit of a catch-all term that we use to refer to the various events that likely occurred to produce established genetic losses in class II PRPS-encoding genes from genomes or non-genome duplication gains in PRPS-encoding genes, as we observe in the genome of *Erpetoichthyes* where two duplicated *prps1-like* genes are located next to *prps1* (see Supplementary Figure S2). These might be meiotic homologous recombination-dependent, or through other mechanisms. For example, reviewer 2's astuteness already identified examples in Figure 1B, D, and E where strict exon-intron gene structure is not maintained, which implicates genetic reshuffling in some form or fashion.

L447 The method for intron/exon boundary detection lacks sufficient detail. (what does "sourced" mean in this context). Also, need a coma after "available" on line 449.

We thank the reviewer for the opportunity to provide important methodological details and for the correction of the grammatical error. We have now edited the methods section as following:

L459-464: "For splicing analysis, genomic DNA or mRNA sequences used to determine splice site junctions for each PRPS homolog were obtained from the NCBI and Ensembl databases. When exonic and intronic sequences were not explicitly provided, we used homologous sequences to predict amino acids and applied splice junction rules (canonical GT/AG and non-canonical GC/AG) to define exon-intron boundaries. The corresponding exonic and intronic sequences for each identified homolog used in the analysis are available on figshare."

Comma has been added after "available"

L464: "available,"

Minor issues:

Throughout the text the authors use "highly homologous" to mean "highly similar" or maybe "high sequence identity". Genes/sites/positions are either homologous or not; homology is a state not a continuous trait.

The reviewer is correct to point out our sloppiness. We have made appropriate corrections to the text.

L42: "highly homologous" changed to "*homologous*"

L87: "greater amino acid sequence homology" changed to "*greater amino acid sequence identity*"

L116: "strong homology" changed to "*homology*"

L159: "greater amino acid sequence homology" changed to "*greater amino acid sequence identity*"

L420 delete "its"

Noted, and done. Thank you for catching this!

RESPONSE TO REVIEWERS' COMMENTS

Reviewer #2 (Remarks to the Author):

The authors have done an excellent job addressing the concerns raised in my initial review. They have clarified key points regarding the inheritance of Class II PRPS enzymes, improved transparency through expanded and revised tables and figures, and appropriately tempered previous overstatements about selective pressures and adaptation. I particularly appreciate the nuanced revisions that acknowledge the limitations of their data and incorporate relevant literature, including the connection to Ohno's 2R hypothesis (though I note Ohno's original work remains uncited, why?) and broader trends in retroduplication.

The manuscript is now significantly clearer and more rigorous, particularly in its treatment of evolutionary inference and gene family expansion. The authors' responses are thoughtful and thorough, and the revised manuscript represents a valuable contribution to our understanding of PRPS gene evolution in eukaryotes. I am satisfied that my concerns have been fully addressed.

We thank Reviewer 2 for their kind words and for their much-appreciated effort and thoughtfulness that went into improving our manuscript. We also thank the reviewer for noting our omission of Ohno's original work. We have now addressed this by citing Ohno's seminal work as Reference #32.